# Combinatorial allosteric modulation of agonist response in a self-interacting G-protein coupled receptor

Marco Patrone [iD] [1], Eugenia Cammarota[2], Valeria Berno[3], Paola Tornaghi[1], Davide Mazza[2] & Massimo Degano [iD] [1]*

The structural plasticity of G-protein coupled receptors (GPCRs) enables the long-range transmission of conformational changes induced by specific orthosteric site ligands and other pleiotropic factors. Here, we demonstrate that the ligand binding cavity in the sphingosine 1-phosphate receptor S1PR1, a class A GPCR, is in allosteric communication with both the β-arrestin-binding C-terminal tail, and a receptor surface involved in oligomerization. We show that S1PR1 oligomers are required for full response to different agonists and ligand-specific association with arrestins, dictating the downstream signalling kinetics. We reveal that the active form of the immunomodulatory drug fingolimod, FTY720-P, selectively harnesses both these intramolecular networks to efficiently recruit β-arrestins in a stable interaction with the receptor, promoting deep S1PR1 internalization and simultaneously abrogating ERK1/2 phosphorylation. Our results define a molecular basis for the efficacy of fingolimod for people with multiple sclerosis, and attest that GPCR signalling can be further fine-tuned by the oligomeric state.

---

[1] Biocrystallography Unit, Division of Immunology, Transplantation, and Infectious Diseases, IRCCS San Raffaele Scientific Institute, Milan, Italy. [2] Center for Experimental Imaging, IRCCS San Raffaele Scientific Institute, Milan, Italy. [3] Advanced Light and Electron Microscopy Bioimaging Center ALEMBIC, IRCCS San Raffaele Scientific Institute, Milan, Italy. *email: degano.massimo@hsr.it

The G-protein-coupled receptor (GPCR) seven-helix transmembrane module represents one of the most successful protein folds in living organisms, demonstrated by the spreading and diversification of GPCR-coding genes across eukaryotic evolution[1], even transferred to some viruses[2]. The architectural plasticity of the membrane-spanning hydrophobic helical bundle parallels a remarkable structural flexibility in transducing signals, making GPCRs versatile allosteric proteins[3]. Indeed, internal cavities and steric micro-switches enable the attainment of a wide range of specific ligand-induced receptor conformations at the cytosolic side[4] that affect dynamically the recruitment of the downstream mediators, and thus the transduction outcomes[5]. The interplay between heterotrimeric G-proteins and arrestins for the receptor occupancy modulates the selection and the activation kinetics of intracellular signalling effectors[6]. While the Gα proteins apparently activate subunit-specific effector pathways, the activity of the two non-visual β-arrestins (β-arrestin-1 and β-arrestin-2, also called arrestin-2 and arrestin-3, respectively) has emerged more pleiotropic and flavoured by the quality of their interaction with the receptors[7]. Indeed, GPCR:β-arrestin interaction can result in several outcomes, ranging from the classical receptor desensitisation of the agonist-induced Gα-mediated signal pathways, up to providing multiprotein signalling scaffolds[8]. Recruitment of β-arrestins by an active state-GPCR mainly leads to either a loose binding to the receptor cytosolic tail, or a tight simultaneous interaction with the receptor core and tail[9], albeit recent observations suggest that β-arrestins can also directly bind the receptor core only[10]. While tightly bound arrestins shield the receptor core cleft that accommodates the Gα subunit[11,12], the loosely bound conformation is compatible with the co-binding of the G protein heterotrimer[13]. Multiple tail and intracellular loop 3 phosphorylation patterns[14,15] seem to play a pivotal role in determining the mode and the strength of the binding in β-arrestin:GPCR complexes, and GPCRs have been classified based on the stability of their binding to β-arrestins[16]. Those patterns are the products of the five non-retinal GPCR kinases (GRKs) whose possible preference for different agonist-induced GPCR conformers contributes to the balance between G protein and β-arrestin pathways.

The members of the lipid-specific sphingosine 1-phosphate receptor family (S1PR1–5) are the subjects of intense translational research[17]. Indeed, S1PR1 plays a prominent role in immunity, inflammation, and in the vascular system[18], and has been successfully targeted by several agonist compounds with disease-modifying activity[19]. S1PR1 signals through the Gαi protein, inhibiting adenylate cyclase and activating PLCβ, PI3K-Akt, the MAPK pathway and cytoskeleton reorganisation[20]. On the other hand, the contribution of the β-arrestin axis to the S1PR1 signalling has not been so far elucidated. S1P is at relatively high concentration in the blood plasma (0.8–1 μM), while its concentration drops by 100-fold in the peripheral lymph nodes as the consequence of the high S1P lyase activity. Such a gradient is essential for the proper homing of B and T lymphocytes between lymphoid tissues[21], with GRK2-mediated S1PR1 phosphorylation required to prevent lymphocytes persistence in the bloodstream[22]. Hence, high haematic S1P tone may act by desensitising lymphocytes, enabling them to exit the blood vessels. Phospho-fingolimod (FTY720-P), the active form of the orally administered pro-drug fingolimod[23], is a S1PR1 agonist that acts as immunosuppressant by inducing a deep lymphopenia[24] and retention in lymph nodes of myelin-autoreactive T cell clones that is beneficial for people with multiple sclerosis (MS). Given its broad effect on T lymphocytes, its use is under investigation for the treatment of other autoimmune and inflammatory diseases[19]. The hallmark of

FTY720-P and other S1PR1 therapeutic agonists is the persistent internalisation of the receptor which makes cells refractory to plasma S1P, drawing an intriguing parallel with the physiological downregulation of the receptor during T cell differentiation via a direct interaction with CD69[25]. Nevertheless, both the signalling potency and the kinetic off-rate constant showed by FTY720-P are like those of S1P on S1PR1[26,27] whereas both fingolimod and FTY720-P plasma concentrations are far lower than S1P, even at high dose regimens[28]. Therefore, S1PR1 therapeutic downmodulation by FTY720-P is caused neither by a super-agonistic binding nor by the disruption of the agonist gradient, while at high concentrations FTY720-P showed a long-lasting in vitro desensitisation of partially active S1PR1[29,30]. Alternative hypotheses require that the FTY720-P:S1PR1 complex is structurally biased toward the receptor desensitisation. Still, at the present, a precise understanding of the molecular events and structural determinants that mediate the immunomodulatory action of FTY720-P is missing.

Here we show that FTY720-P activates S1PR1 to promote a tight binding interaction to β-arrestins, negatively modulating the scaffolding activity of the latter for MAPK proteins. This unique effect of the active form of fingolimod is dependent on S1PR1 oligomerization mediated by the receptor α4 helix, and causes a short-lived Gαi-mediated intracellular response that differs from that of the endogenous agonist S1P. Our results demonstrate that GPCR oligomerization can further modulate the signalling properties of the receptor, stabilising selective conformations at the intracellular side, and expanding the plethora of possible outcomes following orthosteric site occupancy. These additional factors widen the landscape for the design of GPCR ligands with highly specific pharmacological properties.

## Results

**Different kinetics in FTY720-P-mediated and S1P-mediated signalling.** We monitored the phosphorylation of ERK1/2 to assay the relative contributions by Gαi and β-arrestin downstream to S1PR1 activation, since both these mediators can activate the MAPK pathway[4]. First, we established HEK293 cells stably expressing a full-length S1PR1 or S1PR1-eGFP proteins (Supplementary Fig. 1). To avoid the carrier-dependent bias observed in the S1P-S1PR1 axis[31], all the cell-based experiments were performed in culture media complemented with S1P-depleted serum, not perturbing the balance of the serum S1P transporters and keeping lipoproteins available to cells. Stimulating the cells with either S1P or FTY720-P induced an early pERK1/2 peak after 5 min (Fig. 1). Both ligands triggered ERK1/2 phosphorylation through the heterotrimeric G protein branch, with the response made undetectable upon Gαi inactivation by pertussis toxin. Despite the similarity in the magnitude of the early phase and its dependency on Gαi, the S1P-induced and FTY720-P-induced signals diverged in the late response phase. The decay of the S1P-induced signal was slow, with pERK1/2 still detectable at 50% of its maximal value after 60 min. In sharp contrast, FTY720-P induced a pulsed response that rapidly decreased to the baseline level after 20 min, displaying a $t_{1/2}$ of about 10 min. Upon inhibition of GRK1, GRK2, and GRK5 (Fig. 1 and Supplementary Fig. 2A), the pERK1/2 kinetics lacked the early spike of S1PR1 activation for both agonists, indicating that the basal kinase activity is required to achieve the maximal receptor response. The inhibition of receptor phosphorylation also led to a persistent signal with a similar magnitude by both S1P and FTY720-P. The late S1P-induced signal was not affected by the GRK inhibition, while the FTY720-P-induced ERK1/2 phosphorylation was enhanced to the level obtained with S1P. Tagging S1PR1 with eGFP did not interfere with the above ligand-specific kinetics

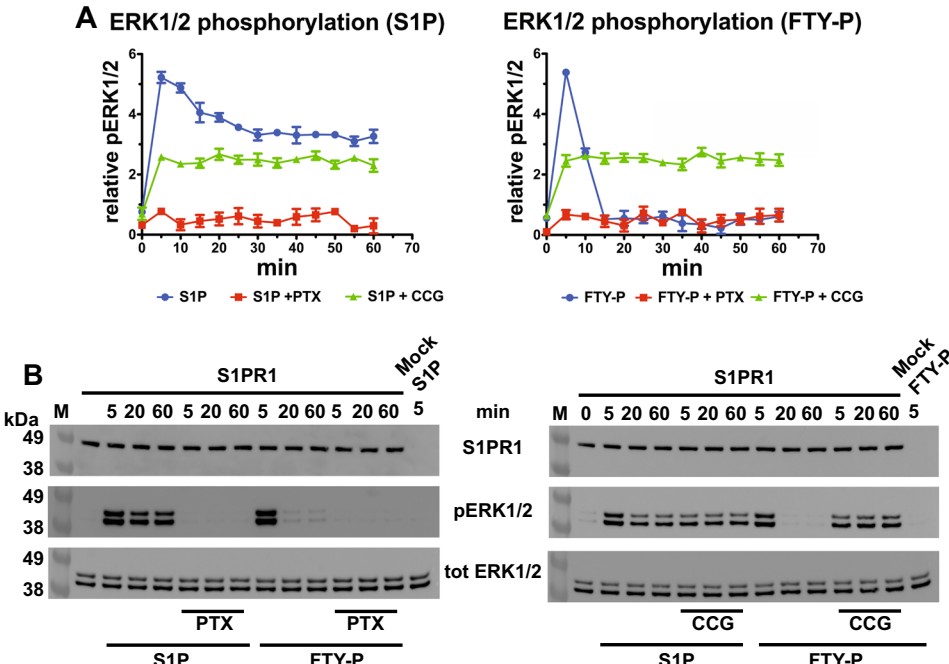

**Fig. 1 Signalling kinetics elicited by S1P or FTY720-P. a** HEK293 cells expressing S1PR1 were S1P-starved for 24 h, with or without 300 ng/mL pertussis toxin (PTX) for 18 h and with 0.1% DMSO alone or with 50 μM CCG215022 (CCG) for 30 min prior of being stimulated with 0.1 μM S1P or FTY720-P for the indicated times. pERK1/2 was measured in the cell extracts by ELISA and expressed as relative units; n = 3 independent experiments; hereafter, ELISA time points are given as mean values ± SEM. **b** Protein extracts from cells treated as in **a** for the indicated times were assayed by immunoblot for S1PR1, ERK1/2 and pERK1/2.

and dependencies of the signalling response (Supplementary Fig. 2B, C), which was in turn abrogated by pre-treating cells with the S1PR1-specific antagonist ML056 (Supplementary Fig. 2D). These observations highlight a clear difference of the kinetic signalling profile induced by the immunosuppressive drug FTY720-P from that by the natural agonist S1P. Moreover, the data indicate that S1P behaves as a Gα$_i$-biased agonist leading to a relatively long-lasting receptor activity, while FTY720-P triggers a rapid and robust, GRK-dependent S1PR1 shut-down.

**α4 helix-mediated S1PR1 oligomerisation at the cell surface.** The role of receptor oligomerisation in GPCR signalling is highly debated. In the crystals of a C-terminally truncated S1PR1 bound to the antagonist ML056[26], two receptor molecules related by a crystallographic two-fold symmetry axis interact in a parallel arrangement via their α4 helices. An extensive buried surface (790 Å$^2$) and a π–π stacking interaction via the side chain of Tyr19 stabilise this dimeric structure (Supplementary Fig. 3, computed $\Delta G_{\mathrm{dim}} = -14.4$ kcal mol$^{-1}$) that is compatible with a physiological arrangement in the plasma membrane (Fig. 2a) with both orthosteric site openings oriented on the same side of the helical bundle. Hence, we investigated whether the oligomeric state of S1PR1 may sustain the differential signalling induced by S1P and FTY720-P. To investigate whether the dimerisation surface observed in the constructs used for the x-ray structure determination also mediates S1PR1 self-association, we engineered a triple mutant (mS1PR1) to destabilise the interface by replacing Tyr19, Phe161[4.43], and Met180[4.62] with Ala residues (Fig. 2a and Supplementary Figs. 1, 3, 4A). The detergent-purified S1PR1 bound the S1P and FTY720-P agonists with nanomolar affinities, with $K_D$ 8.7 ± 0.7 and 5.2 ± 0.3 nM, for S1P and FTY720-P, respectively. The affinity of mS1PR1 for the two agonists was only marginally reduced compared to the unmutated receptor (S1P 21.0 ± 0.8 nM and FTY720-P $K_D$ 14.3 ± 0.9 nM,

Fig. 2b). Consistently, S1P and FTY720-P displayed ten-fold decrease of their potency in stimulating ERK phosphorylation within mS1PR1-expressing clones. On the other hand, the maximal response to the ligands was not affected by the three amino acid substitutions (Fig. 2c), demonstrating that the mS1PR1 retains full capabilities in interactions with intracellular signalling partners. When the S1PR1-eGFP expressing clones were transfected with an unmodified S1PR1 construct, the two proteins co-immunoprecipitated using an anti-eGFP nanobody (Fig. 2d and Supplementary Fig. 4B). Thus, S1PR1 forms oligomers in human cells. No co-immunoprecipitation of the receptors was observed in HEK293 cells stably expressing mS1PR1-eGFP co-transfected with mS1PR1, demonstrating that the residues in the S1PR1 α4 helix participate in oligomerization. Dynamic light scattering measurements (DLS) on the purified, recombinant S1PR1 variants showed a consistent reduction in the size of the micelle-embedded receptor assemblies upon mutation of the α4 helix residues (Supplementary Fig. 4A, C), independently of the detergent system used (Supplementary Fig. 4D), hence confirming that S1PR1 oligomers are the consequence of direct receptor self-interaction. Notably, mS1PR1 retained the ability to form heterotypic oligomers with the wild type counterpart, both in cells and in purified form (Fig. 2d, Supplementary Figs. 4C and 5). Thus, S1PR1 displays a quaternary structure that is mainly supported by non-polar steric matching between α4 helix residues, in line with what shown for the isoform S1PR3[32]. DLS measurements also showed that the binding of FTY720-P specifically increases S1PR1 polydispersity, while it has no effect on mS1PR1 (Fig. 2e, Supplementary Figs. 4E and 6), indicating a ligand-induced wider distribution of the S1PR1:FTY720-P binary complex that is dependent on S1PR1 oligomerization and is reminiscent, albeit on the quaternary structure level, of the conformational heterogeneity observed in other agonist-bound GPCRs[33,34].

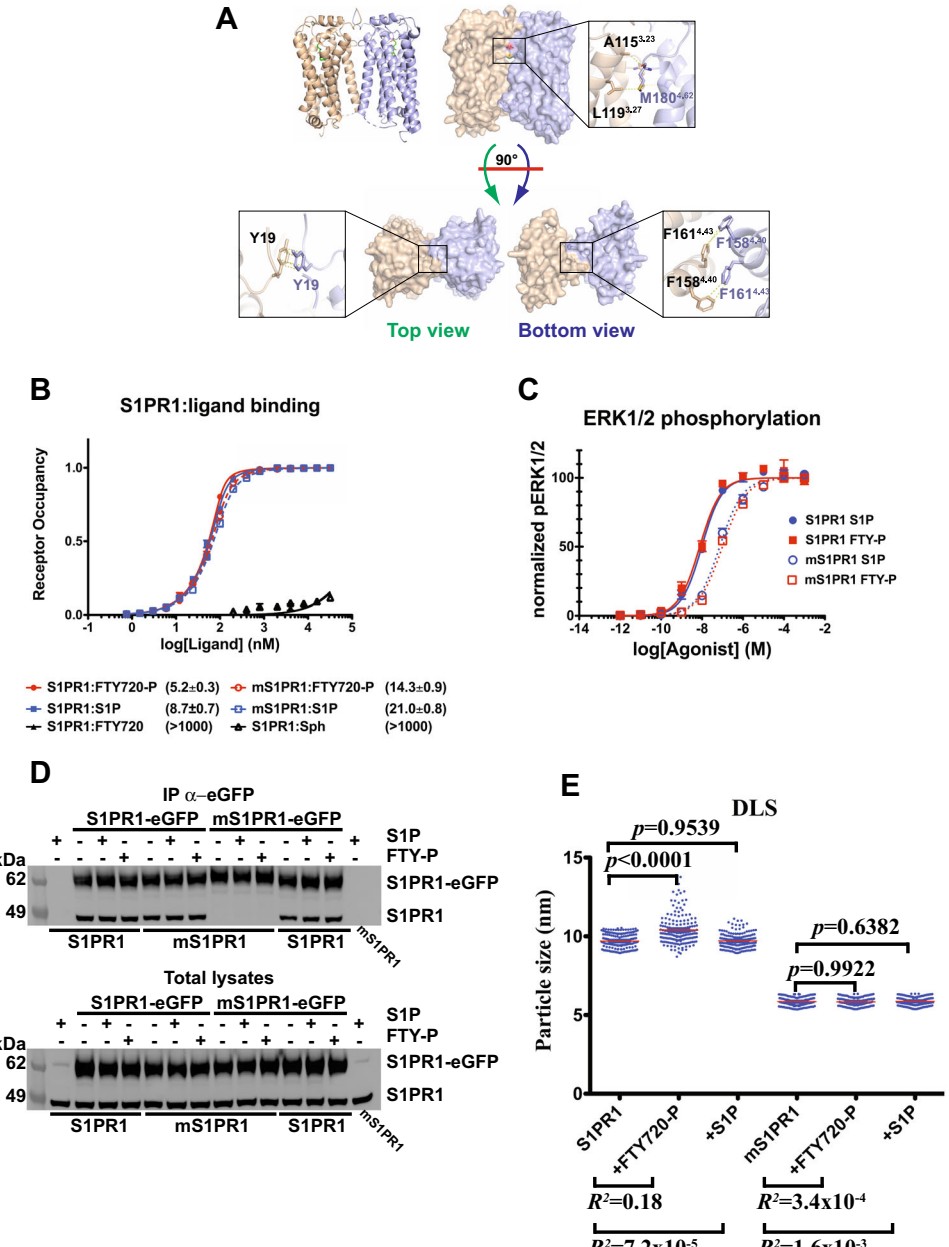

**Fig. 2 Evaluation of the S1PR1 quaternary arrangement. a** Dimer model obtained by PISA software analysis of the crystallographic X-ray structure of human S1PR1 (PDB ID: 3W2Y) with the two protomers in wheat and light blue; the close-up views show the interactions of Met180[4.62], Phe161[4.43], and Tyr19 from one molecule with the counterpart residues Ala116[3.23]/Leu119[3.27], Phe168[4.40], and Tyr19, respectively. **b** Binding curves of S1P or FTY720-P to purified S1PR1 or mS1PR1; 0.1 μM protein was incubated with 0.75–32 μM ligands for 20 min at 37 °C; the binding was measured as the change of the S1PR1 intrinsic fluorescence, normalised to the signal of the 100 μM S1P saturated receptor and expressed as the S1PR1 fractional occupancy. Sphingosine (Sph) and non-phosphorylated fingolimod (FTY720) were as negative controls. $n = 3$ independent experiments. In brackets are the $K_D$ values ± SD (nM) for the tested ligands. **c** S1P-starved HEK293 cells expressing S1PR1 or mS1PR1 were stimulated for 5 min with either S1P or FTY720-P within 1 pM–1 mM range, processed and analysed as in (Fig. 1a); $n = 3$ independent experiments. **d** HEK293 cells expressing S1PR1-eGFP or mS1PR1-eGFP were transfected with plasmids expressing either S1PR1 or mS1PR1; 24 h after transfection, the cells were S1P-starved for additional 24 h and stimulated or not with 0.1 μM S1P or FTY720-P for 5 min; anti-GFP immunoprecipitation (IP) was performed on the cell extracts and the resin eluates analysed by immunoblotting for S1PR1; total lysates were immunoblotted in parallel. **e** Hydrodynamic radii of purified S1PR1 or mS1PR1 in Amphipol 8–35 were measured by DLS at 12.5 μM final protein concentration; individual proteins were incubated for 30 min at 37 °C with the indicated test ligands at 150 μM prior of being read at the same temperature for 10 min with 10 s reads; the experiments were performed in triplicates and pooled; single reads for each condition are shown in the plot; red horizontal bars indicate the median values. $p$ values refer to the two-tailed unpaired $F$-test of the variance; effect sizes are as $R^2$. $n = 175$ reads. 95% CI are reported in Supplementary Fig. 4.

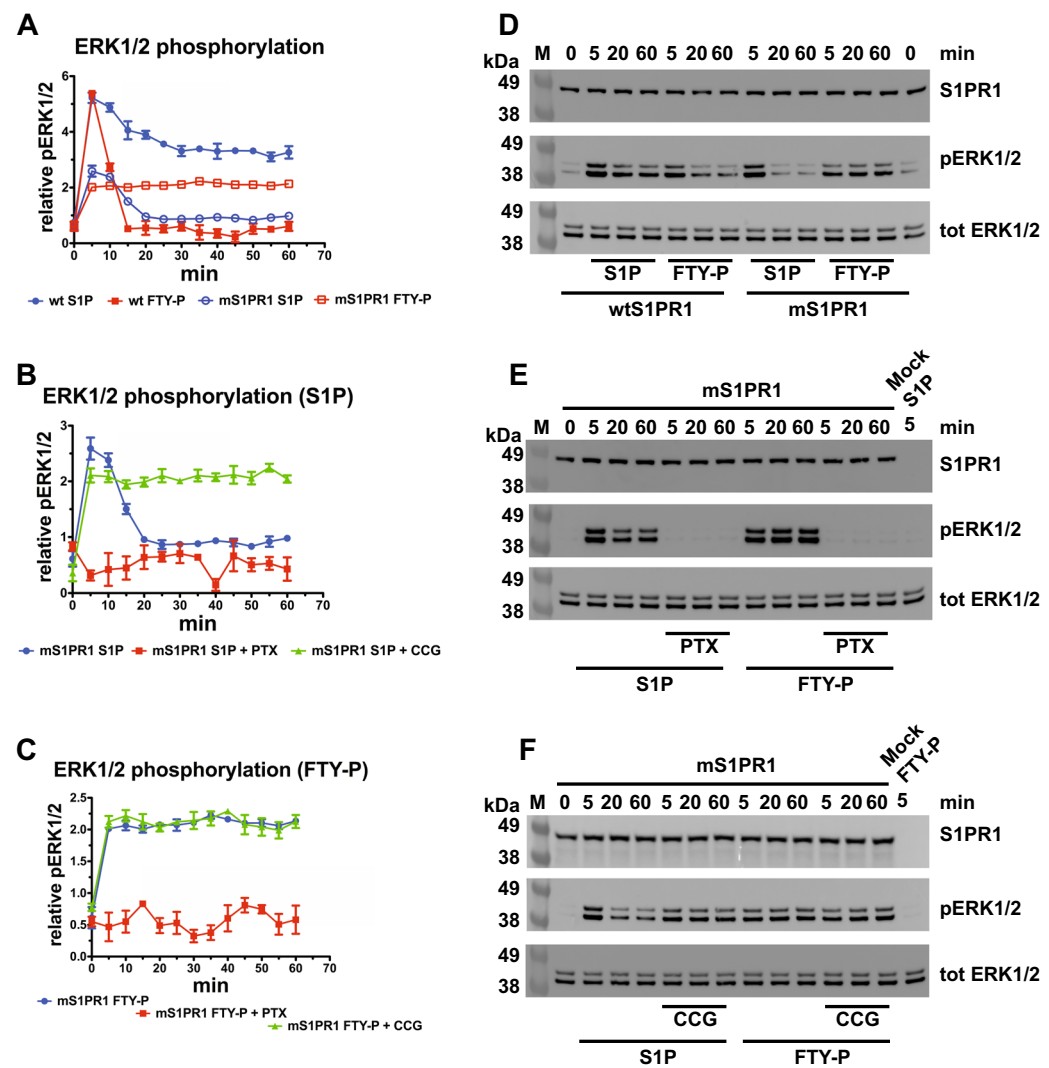

**Fig. 3 S1PR1 oligomerization is a determinant of receptor signalling kinetics.** HEK293 cells expressing mS1PR1 were treated and stimulated as in (Fig. 1a); **a–c** pERK1/2 was measured in the cell extracts by ELISA and expressed as relative units. **a** A comparison with signals from S1PR1 expressing cells is shown; n = 3 independent experiments; **d–f** Cells as in **a–c** have been processed for S1PR1, ERK1/2, and pERK1/2 immunoblotting.

**S1PR1 oligomerization modulates signalling and trafficking**. We explored the effect of S1PR1 oligomerization in response to stimuli by the S1P and FTY720-P agonists. HEK293 cells expressing mS1PR1 displayed a stable level of ERK1/2 phosphorylation upon stimulation with FTY720-P, in sharp contrast to the FTY720-P-induced pERK1/2 kinetics observed with S1PR1 (Fig. 3a, d and Supplementary Fig. 7A, B). At the same time, mS1PR1-stimulated ERK phosphorylation remained Gα$_i$-dependent (Fig. 3b, c, e and Supplementary Fig. 7C) and S1P-induced pERK kinetics was similarly stabilised by GRK inhibition (Fig. 3b, c, f), showing that the mutant receptor maintained the functional relationship with the heterotrimeric G proteins. Hence, mS1PR1 lacks the rapidly decaying profile displayed by the FTY720-P-activated wild type S1PR1. As anticipated by the efficacy analysis, mS1PR1 showed also a weaker transduction efficiency when triggered by S1P, although the pERK1/2 response curve reveals kinetics that are like those of the wild type receptor stimulated by the natural agonist. Thus, the mutations affecting the quaternary assembly of S1PR1 reduce the magnitude of MAPK pathway activation by the natural ligand S1P, and severely modify the FTY720-P specific response.

One of the hallmark events of FTY720-P binding to S1PR1 is the internalisation and persistent downregulation of the receptor,

followed by degradation at longer time points[29,35]. We used confocal microscopy to visualise receptor trafficking of eGFP-tagged S1PR1 and mS1PR1, and to quantitatively assess the differences in internalisation following agonist binding. Binding of FTY720-P to S1PR1, but not of S1P, lead to the rapid (~5 min) observation of large receptor-positive clusters within the cytosol. The difference between S1PR1 and mS1PR1 signalling after FTY720-P stimulation is also paralleled by a decreased propensity to internalisation by the mutant receptor following the binding of the agonist in receptor-expressing cells (Fig. 4a and Supplementary Fig. 8). S1P-stimulated S1PR1 did not undergo substantial internalisation with respect to the basal level observed in untreated cells. Conversely, FTY720-P induced—as expected—a marked endocytosis of S1PR1, with an approximatively threefold increase in the mean number of S1PR1-positive cytosolic vesicles per cell compared to the basal level. On the other hand, mS1PR1 was poorly internalised when stimulated by either S1P or FTY720-P. FTY720-P-stimulated cells showed only a modest increase of the mean number of mS1PR1$^+$ cytosolic vesicles compared to the baseline. Accordingly, only the FTY720-P-stimulated S1PR1 underwent significant receptor degradation (Fig. 4b). In agreement with previous work[35], we observed that S1PR1 was barely detectable in cell lysates after 4 h-long

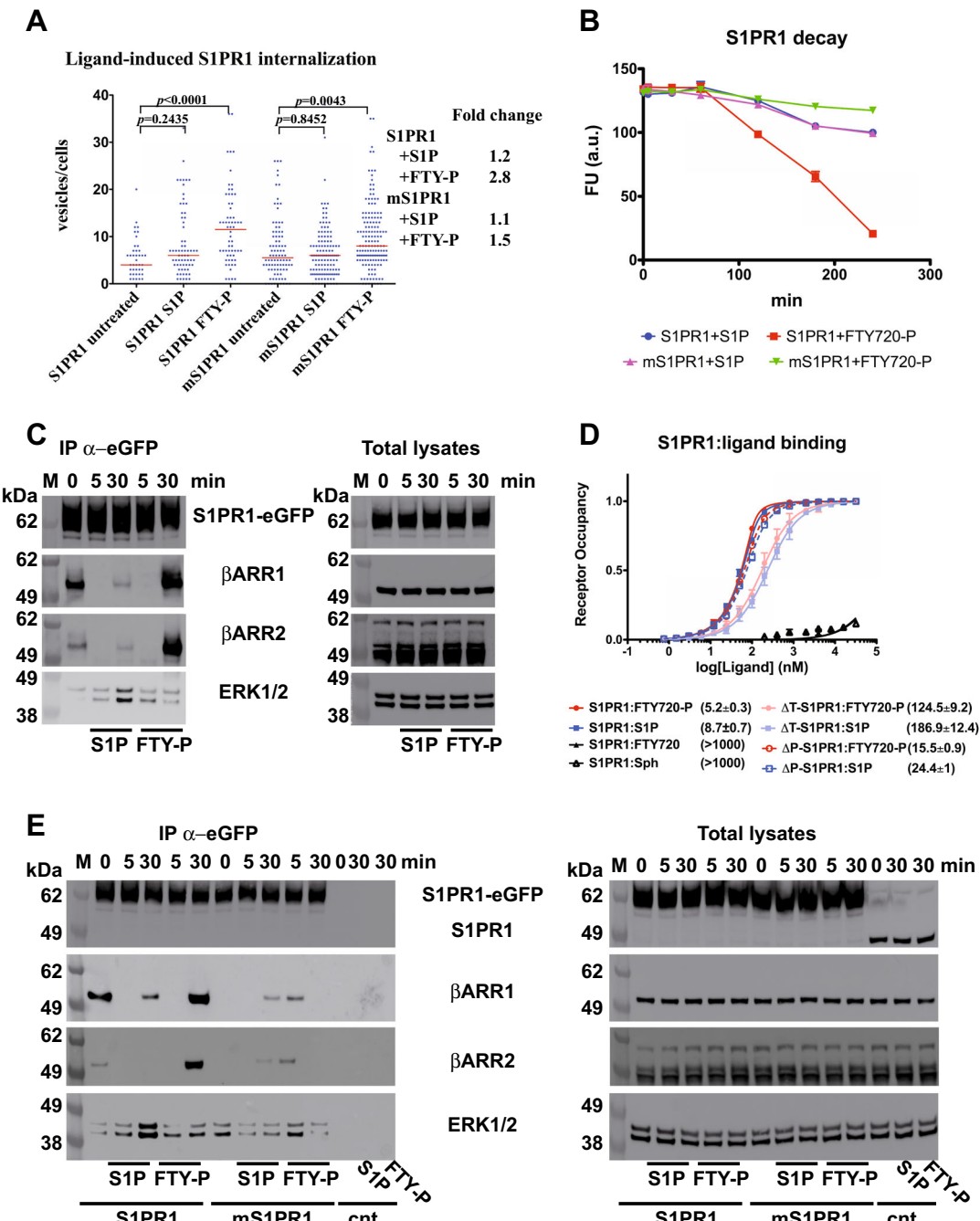

**Fig. 4 S1PR1 interaction with β-arrestins and ERK1/2 is ligand-dependent and oligomerization-dependent. a** S1P-starved HEK293 cells expressing either S1PR1-eGFP or mS1PR1-eGFP were stimulated or not with 0.1 μM S1P or FTY720-P for 20 min; cells were live imaged under confocal microscopy and non-membrane eGFP+ cluster were automatically counted from the cell mid-planes; the data set consisted of three independent experiments for each condition whose outputs were pooled for the analysis; the plot shows the number of clusters/cell and the median values; p values refer to the Kruskal-Wallis two-tailed unpaired ANOVA test. Cohen's d-values for S1PR1: untreated vs. S1P = 0.5, untreated vs. FTY-P = 1. Cohen's d-values for mS1PR1: untreated vs. S1P = 0.13, untreated vs. FTY-P = 0.35. S1PR1 untreated $n = 42$ cells, 95% CI = 4-6.6; S1PR1 S1P $n = 69$ cells, 95% CI = 6.7-10.2; S1PR1 FTY-P $n = 62$ cells, 95% CI = 9.9-14.1; mS1PR1 untreated $n = 90$ cells, 95% CI = 6.1-8.8; mS1PR1 S1P $n = 128$ cells, 95% CI = 5.7-7.7; mS1PR1 FTY-P $n = 150$ cells, 95% CI = 8.7-11. **b** Cells as in **a** were treated with either S1P or FTY720-P from 0 to 240 min; eGFP fluorescence was read from cell extracts and plotted as fluorescence arbitrary units; $n = 3$. **c** HEK293 cells expressing S1PR1-eGFP were S1P-starved for 24 h and stimulated with 0.1 μM S1P or FTY720-P for 5 or 30 min; anti-GFP immunoprecipitation (IP) was performed on the cell extracts and the resin eluates analysed by immunoblotting for S1PR1, β-arrestin 1 (βARR1), β-arrestin 2 (βARR2), or ERK1/2 (in the βARR1 blot, the marker lane was digitally moved to the left with respect to the original picture, see Supplementary Fig. 10 for the original blot); total lysates were immunoblotted in parallel. **d** Binding curves of the indicated purified S1PR1 variants were recorded and plotted as in (Fig. 2b) and compared with the S1PR1 curves, $n = 3$. **e** S1P-starved HEK293 cells expressing S1PR1-eGFP, mS1PR1-eGFP or S1PR1 (cnt samples) were stimulated and processed as in **a** and assayed for S1PR1, β-arrestin 1 (βARR1), β-arrestin 2 (βARR2), and ERK1/2.

treatment with FTY720-P, while mS1PR1 levels remained stable upon agonist stimulation. Hence, the hallmark S1PR1 down-regulation that follows FTY720-P binding to S1PR1 requires the native oligomerisation capability of the receptor at the plasma membrane.

**FTY720-P binding to S1PR1 prevents scaffolding by β-arrestin.** The differences in S1PR1 signalling and endocytotic trafficking induced by the S1P and FTY720-P ligands prompted us to investigate variations in the association with intracellular adaptors. In resting cells, unstimulated S1PR1 is involved in a basal interaction with β-arrestin1 and 2 isoforms (Fig. 4c). Immediately upon stimulation by either agonist, S1PR1-associated β-arrestins became undetectable at the time-point of maximal signalling activity. In the late signalling phase, FTY720-P-activated S1PR1 showed a stable binding to β-arrestins while S1P binding induced only a limited interaction. This behaviour parallels what observed in the endocytosis process, where S1PR1 was internalised much less when cells were treated with S1P compared to FTY720-P. Strikingly, β-arrestin recruited more ERK1/2 onto the S1P-activated S1PR1 than the stable S1PR1:β-arrestin complex induced by FTY720-P, which did not show an increase of the interaction with ERK1/2 above baseline levels. These observations closely parallel the differences in signalling that discriminate the natural agonist S1P from the immunosuppressant drug FTY720-P, and demonstrate the antonymic relationship between β-arrestin binding strength and the capability of recruiting ERK1/2 by the activated S1PR1 to sustain a long-lasting signalling response. Indeed, the inversed proportionality of the β-arrestin: ERK1/2 co-recruitment onto S1PR1 clearly shows that the two agonists promote distinct modes of receptor:β-arrestin interaction. We investigated whether a molecular connection existed between the S1PR1 orthosteric site and the C-terminal tail that is known to be involved with interactions with β-arrestins (Fig. 4d). The removal of the S1PR1 carboxy-terminal amino acids encompassing one of the the β-arrestin-binding regions (ΔT-S1PR1) drastically reduced the binding affinity (S1P 186.9 ± 12.4 nM and FTY720-P $K_D$ 124.5 ± 9.2 nM), while mutation to alanine of the S-acylated Cys residues[36] downstream of helix α8 (ΔP-S1PR1) had a limited effect ($K_D$ 24.4 ± 1.0 nM for S1P and 15.5 ± 0.9 nM for FTY720-P). The relative affinity towards FTY720-P and S1P is maintained in all engineered S1PR1 constructs, thus the mutations introduced do not result in a structural bias of the receptor towards one of the ligands. Hence, the S1PR1 cytoplasmic tail that interact with β-arrestins communicates with the orthosteric site across the receptor architecture, having a major effect on the affinity towards both ligands. It is thus conceivable that different S1PR1 agonists may induce different C-terminal conformations, through the here described allosteric communication, with a selective effect on β-arrestin association.

**S1PR1 oligomers modulate β-arrestin scaffolding activity.** We further asked the question whether S1PR1 oligomerization bolstered the ligand-specific modulation of the β-arrestin scaffolding activity. A comparison of the β-arrestin association kinetics to either wild-type or mS1PR1 demonstrated that the perturbation of the receptor quaternary structure severely compromised the ligand-specific modulation of β-arrestin activity (Fig. 4e). Indeed, neither S1P nor FTY720-P were capable to induce a stable interaction of the mutated mS1PR1 with β-arrestin or, following endocytosis, receptor degradation. Moreover, ligand stimulation did not increase the amounts of ERK1/2 co-precipitated with mS1PR1. The significantly reduced receptor:β-arrestin association in unstimulated cells is in line with the diminished maximal

ERK1/2 phosphorylation recorded for mS1PR1 (Fig. 3) which suggests that S1PR1 oligomerization is also required for both the basal and peak activities of the receptor.

Ligand-specific differences in receptor-associated β-arrestins were still observable in mS1PR1-expressing cells. Indeed, paralleling the signalling kinetics curve, S1P-activated mS1PR1 followed the same time profile of β-arrestin interaction as the wild type receptor but with a lower efficiency that matched the absence of a detectable stimulation of the ERK1/2 interaction. Instead, after an initial weak and transient mS1PR1:β-arrestin binding, FTY720-P-activated mS1PR1 did not associate with β-arrestin at later time points. Hence, the absence of the tight β-arrestin-binding explains the failure to induce the hallmark late signalling shut-down by FTY720-P in cells expressing the mutated receptor.

Our data show that a stronger β-arrestin binding corresponds to poor ERK1/2 scaffolding. We took advantage of the deposited crystal structure of the tight rhodopsin:visual arrestin complex[37,38] to model the possible structural consequences of receptor dimerisation on β-arrestin signalosome assembly. Superposition of two arrestin-bound rhodopsin, which interact with the receptor core (Fig. 5a), onto the S1PR1 crystallographic dimer indicates that the ERK1/2 interaction surface of β-arrestin[39,40] would be close to a second adaptor molecule, and thus sterically hindered in recruiting stoichiometric quantities of the MAP kinase. Coherently, mS1PR1 is impaired in transmitting the FTY720-P shut-down signal. Thus, the homology modelling supports a steric control by S1PR1 dimers of the association of ERK1/2 onto β-arrestin molecules bound in the tight, receptor core engaging conformation. On the other hand, the efficient scaffolding activity elicited by S1P stimulation requires the endogenous agonist to promote a different mode of association of β-arrestin, perhaps mainly with the receptor tail (Fig. 5b). This is supported by the classification of S1P-stimulated S1PR1 as a class A GPCR with transient β-arrestin binding[7]. Although our data do not allow to infer the binding mode of β-arrestin to mS1PR1, our results are consistent with a model where impairment of oligomerization correlates with the general absence of S1PR1 competence for a stable interaction with β-arrestin.

## Discussion

The tertiary structure of Class A GPCRs is stabilised mainly by hydrophobic packing within the membrane bilayer. The few inter-helical polar bonds are not sufficient to rigidify the fold, and have been proposed to act as flexible joints to promote propagation of ligand-induced structural rearrangements across the molecule. The outcome is that GPCRs have deformable architectures whose response to their cognate agonists can be further modulated by a variety of structural factors. Here we uncover yet another GPCR regulating factor by showing that receptor oligomerisation can be crucial for the fine-tuning of agonist-specific effects. Indeed, the previously neglected S1PR1 self-interaction is a crucial molecular determinant in enhancing the receptor plasticity in response to orthosteric site ligands. The engineered mS1PR1 protein binds agonist compounds with reduced affinity, recruits β-arrestins less efficiently, and does not achieve comparable levels of early-phase ERK1/2 phosphorylation. Thus, receptor oligomerization has a buttressing effect on the S1PR1 structure to allow both the attainment of the native binding site structure and the association of β-arrestins in absence of ligand, priming the GPCR for a full response to agonists.

The differences in the functional outcome upon agonist binding to S1PR1 are not caused by a higher receptor affinity of FTY720-P over S1P (Figs. 2, 4), in agreement with their comparable signalling potencies (Figs. 1, 3 and ref. [26]). At the same time, truncating the C-terminal segment or—to a lesser

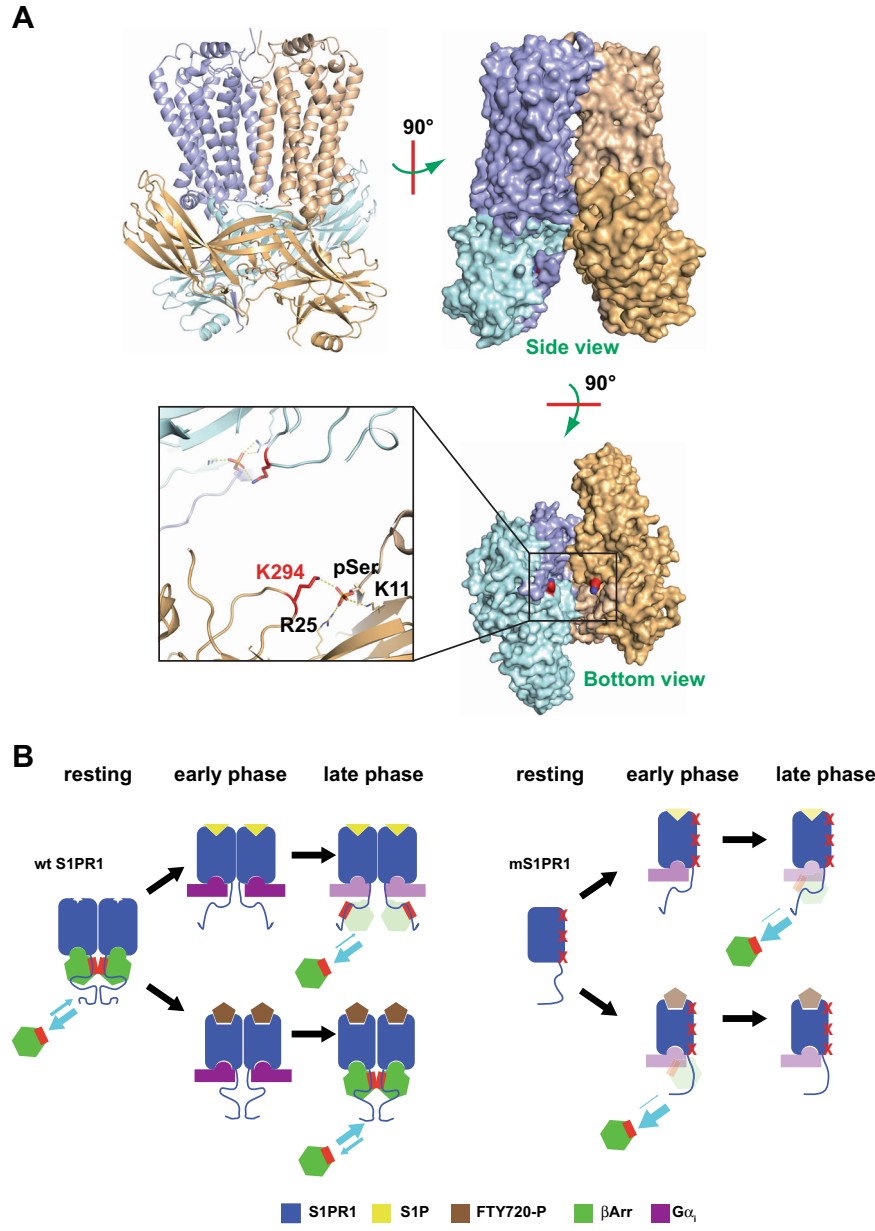

**Fig. 5 Modulation of the S1PR1 signalling by receptor oligomerization. a** Model of dimeric S1PR1:arrestin tight complex; the model was built by aligning the S1PR1 crystallographic dimer (PDB ID: 3V2Y, Fig. 3) with the X-ray rhodopsin:visual arrestin complex structure (PDB ID: 5W0P); the colour code of the S1PR1 protomers is the same as in (Fig. 3), while the two arrestin monomers are coloured in pale cyan and light orange, respectively; the rhodopsin phosphorylated C-tail was left in the model to show the steric hindrance and the electrostatic bond network engaging the arrestin K294 residue—critical for the β-arrestin:ERK1/2 interaction—with the phospho-C-tail; the same bond arrangement is present in the complex β-arrestin 1:V$_2$Rpp peptide (PDB ID: 4JQI); the residue numbering refers to the human β-arrestin 1 protein sequence. **b** Schematic of the tripolar allosteric connection in S1PR1; the oligomeric wild type S1PR1 has a basal interaction with β-arrestins, predisposing the receptor for the agonist binding; during the early phase response, the heterotrimeric G protein effectively squelches β-arrestins from S1PR1, leading to the maximal signalling response; in the S1P-induced conformation, a loose S1PR1:β-arrestin complex is expected to accumulate, with a weak propensity to compete with the Gα$_i$:receptor core interaction, and supportive of the β-arrestin scaffolding activity (the red side in the β-arrestin symbol stands for the ERK1/2-interacting surface); the FTY720-P binding transfers a different signal to the C-tail in the oligomeric S1PR1, leading to a late phase where the tightly bound β-arrestin turns off the Gα$_i$-dependent signalling and the ERK1/2 recruitment is hampered by the oligomeric receptor arrangement; the disruption of the S1PR1 self-interaction reduces the abundance of the basal receptor conformers that favour the agonist binding, and weakens the transduction from the engaged orthosteric site to the S1PR1 cytosolic side; moreover, an altered communication causes the lack of the FTY720-P-induced strong β-arrestin binding. Gα$_i$ and β-arrestin are represented overlapped to account for the uncertainty on the binding mode by β-arrestin onto mS1PR1. The colour intensity of both ligands and mediators are to indicate the respective binding site occupancy.

extent—preventing its S-acylation reduced the capacity of S1PR1 to bind either S1P or FTY720-P (Fig. 4d). This feature unveils an unprecedented allosteric connection between the C-tail and the orthosteric binding site within the flexible architecture of S1PR1 and extends the knowledge on the conformational flexibility of helix 8 and its relationship with arrestin function[41]. Thus, the ligand binding site of S1PR1 is not only in a reciprocal allosteric relationship with the cytosolic core cleft[42] to trigger G-protein and β-arrestin association, but can likewise propagate conformational changes to the very cytosolic end of the receptor that is target for GRK phosphorylation and β-arrestin binding. This finding rationalises how the two different agonists with strikingly similar biochemical features induce a differential S1PR1-mediated transfer of information to the β-arrestin cascade (Fig. 5b). We argue that the bidirectional structural dependence linking the orthosteric site and the cytosolic tail of S1PR1 may determine distinct conformations of the S1PR1 C-tail from different modes of ligand site occupancy. This speculation reconciles the so far contrasting observations reporting two different phosphorylation patterns associated to S1P and FTY720-P, both dependent on GRK2. Five membrane proximal Ser residues (5S) are phosphorylated upon FTY720-P stimulation and are required for the fingolimod-induced lymphopenia[43]. At the same time, the so called TSS motif in the distal portion of S1PR1 tail is phosphorylated following the S1P stimulus, and is required for the S1PR1 desensitisation by the high-tone haematic S1P[22]. The phospho-pattern locations along the GPCR C-tail have been recently related to their ability of inducing a tight binding-proficient conformation of β-arrestins[44]. Membrane-proximal patterns are better oriented to unlock the β-arrestin finger loop that inserts into the receptor core cleft in the GPCR:β-arrestin tight configuration, which also account for the role of the membrane-engaging β-arrestin moiety in the tight binding[45]. Our results fit well this picture, since the S1P:S1PR1 complex displayed a modest propensity in recruiting β-arrestins and undergoing internalisation and degradation, while FTY720-P induced a stable S1PR1:β-arrestin interaction leading to deep receptor internalisation and subsequent down-modulation (Fig. 4a, b, and ref. [31]). It is worth to note that S1PR1 undergoes a physiological downregulation during naïve-to-effector T lymphocyte differentiation via an interaction with CD69[25] that requires S1P-induced Gα$_i$ signalling, the proximal 5S motif in the S1PR1 cytosolic tail, and the receptor α4 helix.

The biochemical role of oligomerization in Class A GPCRs is a long-debated issue. Several rhodopsin-like GPCRs (β$_2$AR[46], β$_1$AR[47], CXCR4[48], H1R[49], κOR[50], μOR[51]) have been crystalized as parallel homodimers, although not one single surface of the helical bundle appears as a preferential side for the dimerisation of GPCRs. We noticed that S1PR1 was arranged in the orthorhombic crystals as symmetric homodimers through residues at the α4 helix surface (Fig. 2), analogously to what demonstrated for S1PR3[32]. Structure-based[26] mutation of critical residues in the putative S1PR1 interaction surface (Supplementary Fig. 3) allowed us to engineer the mS1PR1 receptor variant that lost both the self-interaction ability and, more importantly, the FTY720-P-induced bias to the inhibiting β-arrestin activity (Figs. 3, 4a). This result also raises the intriguing possibility that S1PR isoforms co-expressed in several cell types (e.g., S1PR1 and S1PR3 are both expressed on the surface of cardiomyocytes) may form hetero-oligomers and, in turn, modulate the intracellular signalling in a district-specific fashion through the selection of oligomer-dependent Gα isoforms. Although this hypothesis requires experimental verification, it may for instance rationalise the reported interdependency between S1PR1 signalling and S1PR2/3 expression in mouse ventricular myocytes[52].

Our results demonstrate that the information received from the S1PR1 self-interaction surface modulates the outcome of the ligand-induced β-arrestin activity (Figs. 3, 4). The effect of FTY720-P binding on the oligomer polydispersion in solution is indicative of a ligand-specific modulation of the receptor structure that requires further characterisation. Along with the role played by the receptor C-tail in enhancing the ligand affinity in the recombinant constructs, the evidence points at the existence of a tripolar connection within S1PR1, bridging the self-interaction surface, the ligand-binding site and the cytosolic tail. Such interplay integrates with the specific conformational information carried by different receptor agonists and transmitted to the receptor core. The distinctive impact of FTY720-P on S1PR1 implies that the drug arranges differently from S1P into the S1PR1 ligand pocket—since the respective dissociation rate constants have been reported as similar[27]—ultimately allowing the receptor to attain a specific conformation. Coherently, mS1PR1 still transmitted differently in response to either S1P or FTY720-P (Figs. 3, 4e), and the drug-induced signalling profile resulted critically affected by the absence of oligomerization in mS1PR1. S1PR1 self-interaction is also essential for the basal presence of S1PR1:β-arrestin complex in non-stimulated cells (Fig. 4e), which is required for the maximal activation of the agonist-evoked signalling pathway (Figs. 1, 3), likely by predisposing the S1PR1 orthosteric site to a more efficient interaction with its agonists. At the same time, mS1PR1 failed in activating a detectable S1P-induced β-arrestin scaffolding ERK1/2 (Fig. 4e), reinforcing the evidence that the S1PR1 self-interaction surface communicates with the β-arrestin-interacting receptor elements.

Taken together, S1PR1 oligomerization and ligand-specific effects cooperate to diversify the intracellular signalling response to agonists. The differential signalling kinetic profiles (Figs. 1, 3), endocytic/degradation behaviour of the activated receptor (Fig. 4a, b) and association to ERK1/2 (Fig. 4c, e) indeed suggest a distinct configuration of the S1PR1:β-arrestin complex in response to S1P or FTY720-P binding. The homology model shows that the tight binding to a α4:α4 S1PR1 dimer would obstruct the β-arrestin surface that is required for the ERK1/2 interaction (Fig. 5 and ref. [39]) which is in full agreement with the experimental results on the FTY720-P-evoked signalling. In principle, the reduced amount of β-arrestin onto S1P-activated S1PR1 could be interpreted as an overall lower number of receptor molecules proficient for binding, without implying different modes of interaction. However, a distinct quaternary arrangement of the S1P:S1PR1:β-arrestin ternary complex has to take place to account for the higher level of co-recruited ERK1/2. Hence, the type of interaction that individual GPCRs engage with β-arrestins is not only encoded in the receptor sequence but it is susceptible to be ligand-dependent. Moreover, the multimeric structure of GPCRs, here exemplified by S1PR1, can sustain a multifaceted role of stabilising the signalling-competent conformations, modulating the conformational variability induced by orthosteric site ligands, and influencing the selection of specific mediators on the signalosome.

Finally, our results indicate that the agonist FTY720-P exerts its therapeutic sequestration of the receptor from the cell surface by inducing an oligomer-dependent, α4 helix-mediated down-regulation. Thus, it is tempting to speculate that FTY720-P may be harnessing a physiological mechanism, enabling the self-interacting S1PR1 to undergo the effect that S1P evokes on the S1PR1:CD69 heterocomplex. The possibility offered by the S1PR1 quaternary structure for agonists that are similarly capable to manipulate the connection linking the receptor lateral surface to the β-arrestin binding tail from inside the orthosteric site warrants the further development of effective therapeutics targeting this receptor family, and perhaps other class A GPCRs.

## Methods

**Reagents**. Chemicals of general use were from Sigma Aldrich. Sphingosine 1-phosphate, FTY720 (S)-phosphate and ML056 were from Cayman Chemical Company. CCG215022 was from MedChem Express. Detergents for protein purification were from Anatrace. Pertussis toxin was from Thermo Fisher Scientific. Antibodies used: rabbit poly-clonal anti-S1PR1 (Abcam, cat.n. ab137467, lot GR103634-15, dilution 1:1000); rabbit monoclonal anti-β-arrestin 1 (Cell Signalling Technology, cat. n. 12697, clone D8O3J, lot 1, dilution 1:1000); rabbit monoclonal anti-β-arrestin 2 (Cell Signalling Technology, cat. n. 3857, clone C16D9, lot 2, dilution 1:1000); rabbit monoclonal anti-p42/p44 MAPK (ERK1/2) (Cell Signalling Technology, cat. n. 4695, clone 137F5, lot 21, dilution 1:1000); rabbit monoclonal anti-phospho-p42/p44 MAPK (ERK1/2) (Thr202/Tyr204) (Cell Signalling Technology, cat. n. 4370, clone D13.14.4E, lot 17, dilution 1:2000); rabbit poly-clonal anti-LPAR1 (Abcam, cat.n. ab84788, lot GR57094-1, dilution dilution 1:1000).

**Recombinant protein production**. All the constructs were based on the cDNA sequence of human S1PR1 NP_001391 and were engineered by including two terminal TEV-cleavable tags (N-terminal 6×His tag and C-terminal One-strep tag), and inserting the cysteine-free T4 lysozyme crystalisable fragment between Arg231 and Ala243. ΔT-S1PR1 was engineered truncating the receptor sequence at Met326. In ΔP-S1PR1, Cys328, Cys329, and Cys331 were replaced with Ala residues. mS1PR1 was obtained by introducing the Tyr19Ala, Phe161Ala, and Met180Ala mutations in the S1PR1 protein sequence. The corresponding synthetic coding DNA sequences were obtained from GeneArt (Thermo Fisher Scientific), cloned into BamHI/HindIII-digested pFastBac1 vector and used in the Bac-to-Bac baculovirus expression system (Thermo Fisher Scientific) according to the manufacturer guidelines. High titre recombinant baculovirus were obtained from Sf9 cells maintained in SF900II culture medium (Thermo Fisher Scientific) and used to infect High Five™ cells as previously described[53]. Briefly, cells were infected at $1.8 \times 10^6$ cells/mL with a multiplicity of infection of 2.5. Glucose and glutamine were kept in the medium at 30 and 7 mM, respectively, and N-acetyl-cysteine was added at 0.5 mM final concentration. After 48 h of infection, cells were harvested and processed for protein purification. S1PR1, ΔT-S1PR1, ΔP-S1PR1, or mS1PR1 proteins were extracted from cell membrane pellets in 20 mM HEPES, 10 mM NaHCO₃, 1 mM tris(2-carboxyethyl)phosphine (TCEP), 0.5% (w/v) lauryl maltose neopentyl glycol (LMNG), 0.1% (w/v) cholesterol hemisuccinate (CHS), 5% (w/v) glycerol, EDTA-free cOmplete protease inhibitor cocktail (Roche), 25 U/mL Benzonase® (Sigma Aldrich) pH 7.2. Protein extracts were cleared by ultracentrifugation at $120,000 \times g$ for 1 h, 0.2 μm filtered (Supor Akropak 200, Pall) and applied onto StrepTactin Sepharose resin (GE Healthcare). Resin beads were washed in the Amicon® Pro device (Merck) with 20 mM HEPES, 1 M NaCl, 1 mM TCEP, 0.05% LMNG, 0.01% CHS pH 7.2, 5% (w/v) glycerol and subsequently in 20 mM HEPES, 150 mM NaCl, 1 mM TCEP, 0.01% LMNG, 0.002% CHS, 5% (w/v) glycerol pH 7.2 (hereafter called protein buffer). Proteins were eluted in protein buffer with 2.5 mM desthiobiotin (Merck), cleaved with 1:100 molar ratio AcTEV (Thermo Fischer Scientific) for 16 h at +4 °C, passed over a Ni-NTA resin (Biovision) and the flow-through concentrated on Amicon® Ultra centrifugal filters (Merck) with 100 kDa nominal weight cut-off. Finally, the purified proteins were dialysed against an excess of protein buffer and checked by SDS-PAGE. Mono-dispersity was evaluated by dynamic light scattering (DynaPro, ProteinSolutions) and size exclusion chromatography (SEC) on a Superdex 200 increase column (GE-Healthcare). LMNG-to-Cymal 5 detergent exchange was performed step-wise onto the StrepTactin resin before the tag removal by serial washes with protein buffer containing respectively 0.0025, 0.005, 0.0075, 0.01% (w/v) Cymal 5 at 0.01% (w/v) total detergent concentration. LMNG was replaced with Amphipol 8–35 by mixing Amphipol 8–35: purified protein at 4:1 mass ratio for 3 h at 4 °C and then by adsorbing the detergents onto the Bio-beads SM-2 (Bio-Rad) at total detergent:beads 20:1 mass ratio for further 3 h as above. Excess Amphipol 8–35 was removed by SEC.

**S1PR1 expression in cell cultures and S1P-depletion**. Human embryonic kidney-293 (HEK293, ATCC CRL-1573) cells were routinely maintained in complete DMEM/F12 1:1 mixture (Sigma Aldrich) containing 10% FBS and 0.5 mM N-acetyl cysteine. Mycoplasma detection was performed routinely by checking for cytoplasmic DNA and every six months with Universal Mycoplasma Detection Kit (ATCC 30–1012 K). The stop-less coding sequences of either wild-type S1PR1 or mS1PR1 (without the additional modifications present in the constructs for the protein production) were obtained from GeneArt (Thermo Fisher Scientific) and inserted into the HindIII/BamHI-digested pEGFP-N1 vector (Takara) to express S1PR1- and mS1PR1-eGFP. The pEFGP-N1-S1PR1 or pEFGP-N1-mS1PR1 plasmids were further modified by re-introducing the TAG stop codon at the 3′ end of the S1PR1 open reading frame to express untagged S1PR1 or mS1PR1, respectively, using the primer pair 5′–AAGCTTATGGGGCCCACCAGCGTCCCG-3′ and 5′–GGATCCCTAGGAAGAAGAGTTGACGTTTCCAG-3′. Cell monolayers were transfected with JetPEI™ (Polyplus) and the appropriate plasmid vectors. Transfected cell cultures were passed once a week for three times and then the GFP⁺ cells were sorted in a MoFlo XDP (Beckman Coulter). Sorted cell cultures were expanded and sorted again and the expression of the desired chimeric protein verified by immunoblot. Sorting was performed at the Flow Cytometry Resourc, Advanced Cytometry Technical Applications Laboratory of IRCCS San Raffaele Scientific Institute.

Purified wild-type S1PR1 protein was used to deplete the complete cell culture medium of S1P by reverse dialysis. Two hundred nanomole of S1PR1/L of culture medium were used at 200 μM receptor concentration. Under sterile conditions, purified receptor was first dialysed three times against the serum-free medium. Medium-equilibrated S1PR1 was then incubated 24 h at +4 °C in complete culture medium. The depletion procedure was performed twice for each lot of culture medium. The S1P-depleted medium was 0.22 μm filter-sterilised, stored at +4 °C and used for 24 h S1P-starvation of the cell monolayers in the cell-based assays described. The depleted medium was also used to prepare 1000× stock solutions of each of the two S1PR1 agonists or the GRK inhibitor CCG215022, and the 10× ML056 stock solution used in the cell-based experiments.

**Phosphorylation assay**. HEK293 cells were plated in 96 multi-well plates and grown until 90% confluent. After the S1P-starvation, the cell monolayers were treated as described in the text and then processed using the ERK1/2 (pT202/Y204 + Total) ELISA Kit from Abcam (cat n. ab176660), following the manufacturer guidelines and adding the HALT™ Proteases & Phosphatases inhibitor cocktail (Thermo Fisher Scientific) to the lysis buffer provided with the kit. To quantify pERK1/2 and total ERK, 40 μg and 5 μg of total protein extract were used from each well, respectively. The chromogenic signals were read at the Sunrise Tecan multi-plate reader and expressed as pERK/ERK_Total signal ratio. Parallel experiments were conducted in a 6-well plate format and the pERK1/2 present in the cell extracts visualised by immunoblot whose uncropped images are showed in Supplementary Fig. 10. AKT phosphorylation was quantified by ELISA as above in cell lysates after 5 min stimulation, with the AKT (pS473 + Total) ELISA Kit from Abcam (cat. n. ab176657).

**Fluorescence-based ligand binding assay**. The test ligands were dissolved directly in the protein buffer. Purified S1PR1 variants were diluted at 0.1 μM in protein buffer. The intrinsic fluorescence $F_u$ of each of the unbound receptor variants was measured at 37 °C in the Cary Eclipse Varian spectrofluorometer with 278 nm $\lambda_{Ex}$ and 334 nm $\lambda_{Em}$. The test ligand or the mock dilution was added from pre-diluted stocks and the fluorescence point $F_x$ read after 20 min incubation under constant stirring. Intrinsic fluorescence $F_s$ of the saturated receptors was read diluting the receptor variant in protein buffer containing 100 μM of the test ligand. The receptor occupancy was expressed as $(F_u - F_x)/(F_u - F_s)$. Data were plotted and $K_D$ obtained by non-linear regression with ligand depletion using the GraphPad Prism software. Stoichiometry of S1P: and FTY720-P:S1PR1 was obtained from a saturation experiment using 0.75 μM receptor, as the intersection point between the linear regression lines of the pre-saturation and post-saturation phases, respectively.

**Structural analysis**. The crystal structure of the C-terminally truncated S1PR1 with ICL3 substituted with T4L lysozyme bound to the antagonist ML056 (PDB ID: 3V2Y) was visualised using PyMol (http://www.pymol.org). The dimer coordinates were generated by applying the crystallographic (−X, −Y + 1, Z) symmetry operation of the orthorhombic P2₁2₁2 space group to which the crystals belonged. The interaction surface was analysed using PISA (http://www.ebi.ac.uk/pdbe/pisa). The effect of mutations on dimer stability was assessed using the programme FoldX[54]. The structure was automatically checked for close van der Waals contacts using the RepairPDB module. Every amino acid identified at the dimer interface was mutated into the other 19 amino acids, and the variation in the intermolecular interaction energy was computed for each substitution at both chains simultaneously. As recommended for membrane proteins, the calculations were performed setting the dielectric constant to zero and switching off hydrophobic contributions.

**Dynamic light scattering analysis**. For the analysis of the particle size on the unbound receptors, S1PR1 and mS1PR1 were diluted in protein buffer individually at 12.5 μM or mixed together at the molar ratios indicated in the text. Each sample was incubated for 30 min at 37 °C and then the light scattering measured for 10 min at the same temperature with 10 s reads in the DynaPro (ProteinSolutions) driven with Dynamics v6 software. For the analysis of ligand-induced variation of the particle size distribution, S1PR1 or mS1PR1 was diluted at 12.5 μM in protein buffer alone or with 150 μM of the test ligand and the samples pre-incubated and read as above. Each condition was replicated three times and the triplicated reads pooled for the polydispersion analysis using the GraphPad Prism software. Recorded particle diameters were corrected to account for the detergent-containing buffer viscosity ($\eta_r = 1.5$) measured on a MCR72 rehometer (Anton Paar).

**Confocal fluorescence microscopy and image analysis**. Cells expressing S1PR1-eGFP or mS1PR1-eGFP were plated onto poly-L-lysine-coated 8 well μ-slides (Ibidi) at $3 \times 10^4$ cell/well. The day after plating, cells were S1P-starved for 24 h before treating cells with or without 0.1 μM S1P or FTY720-P. Live cell images were acquired on the an inverted point scanning confocal microscope (TCS SP8; Leica Microsystems) equipped with a 37 °C/5% CO₂ chamber and a HC PL APO CS2 63 × (1.4 NA) objective, using LAS X software (Leica Microsystems). Complete z-stacks were acquired with optimised confocal settings to ensure that oversaturation did not occur. Images were deconvoluted using Huygens Professional Software (Scientific Volume Imaging). Mid-plane z-sections were used for counting cytosolic GFP⁺ vesicles in cells stimulated for 20 min. Confocal imaging was performed at

the Advanced Light and Electron Microscopy Bioimaging Centre of IRCCS San Raffaele Scientific Institute. We used a custom Matlab script (Supplementary Data 1) performing the following image analysis: (i) the membranes were segmented from the image, identifying regions of both absolute and local high intensity (Supplementary Fig. 9A–G) in order to include all the membranes independently of their intensity; (ii) morphological erosion was used to clear small objects and to smooth the edges of the identified membranes (Supplementary Fig. 9H, I); (iii) the region of interest was defined as the region completely enclosed by a membrane (Supplementary Fig. 9J, K); (iv) GFP$^+$ cytosolic vesicles were identified as local intensity maxima (Supplementary Fig. 9L–O) within the region of interest and counted. Experiments were replicated independently three times. The number of vesicles from 40 to 150 cells were counted as detailed in Fig. 4 legend and the statistical analysis performed with the GraphPad Prism software.

**Fluorescence-based analysis of S1PR1 down-modulation.** Cells expressing S1PR1-eGFP or mS1PR1-eGFP were S1P-starved and mock-stimulated or stimulated with the indicated agonists from 0 to 240 min. Cells were washed twice in ice-cold phosphate-buffered saline and lysed in 20 mM HEPES, 150 mM NaCl, 1% Triton X-100, 0.15% Nonidet P-40, 1× HALT™ Proteases & Phosphatases inhibitor cocktail, 1.25 U/mL Benzonase. Lysates were clarified were used and 50 μg of total protein from each sample were read in the Cary Eclipse Varian spectrofluorometer with 488 nm $\lambda_{Ex}$ and 509 nm $\lambda_{Em}$ to measure the eGFP amount.

**Immunoprecipitation.** Cells expressing S1PR1-eGFP or mS1PR1-eGFP were S1P-starved and mock-stimulated or stimulated with the indicated agonists for 5 or 30 min; $10^6$ cells were used for each experimental point. Cells were washed twice in ice-cold phosphate-buffered saline and lysed in 20 mM HEPES, 150 mM NaCl, 1% Triton X-100, 0.15% Nonidet P-40, 1× HALT™ Proteases and Phosphatases inhibitor cocktail, 1.25 U/mL Benzonase. Clarified lysates were incubated for 1 h at +4 °C with GFP-Trap® resin (Chromotek) and extensively washed with lysis buffer. Finally, the immunopurified material was eluted in 4% SDS and analysed by immunoblot. The equal loading was based on the GFP amount reading the samples in the Cary Eclipse Varian spectrofluorometer with 488 nm $\lambda_{Ex}$ and 509 nm $\lambda_{Em}$. In some experiments, cells expressing S1PR1 without the GFP tag were used as control and the equal loading was based on total protein concentration measured with the BCA method (G Biosciences).

**Statistics and reproducibility.** All quantitative measures were performed as independent replicates. All attempts to replicate the data were consistent. Statistical analyses with GraphPad Prism and sample sizes are described in the pertinent figure legends. Two-tailed unpaired tests were applied to statistical significance.

**Reporting summary.** Further information on research design is available in the Nature Research Reporting Summary linked to this article.

## Data availability

NCBI reference sequence NP_001391, Uniprot accession code P21453 and PDB codes 3V2Y (S1PR1 in complex with ML056), 5W0P (rhodopsin:visual-arrestin complex) and 4JQI (β-arrestin 1:V$_2$Rpp peptide) were used in this study. Materials and all other data are available from the first or corresponding author upon reasonable request.

## Code availability

The MATLAB script used for the automated determination of the number of eGFP + 2 intracellular clusters from confocal microscopy images is available in Supplementary Data 1.

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

## Acknowledgements

The authors wish to thank colleagues at IRCCS Scientific Institute San Raffaele for suggestions and critical reading of the manuscript, and Cristina Alamprese for the viscosity measures. This work was supported by grants 2011-R-16 (to M.D.) and 2015-R-18 (to M.P.) from FISM, and PE-2013-02355206 from the Italian Ministry of Health (to M.D.)

## Author contributions

M.D. and M.P. conceived the study; M.P. and P.T. performed all cellular and molecular experiments; M.P. and M.D. analysed the S1PR1 structure; V.B. and M.P. designed and performed the microscopy experiments; E.C. and D.M. designed and performed the analysis of confocal images to quantify the receptor trafficking. M.P. and M.D. draughted the manuscript with contribution from all authors.

## Competing interests

The authors declare no competing interests.
