## [Peer Review File · Communications Biology]

Reviewers' comments:

Reviewer #1 (Remarks to the Author):

The authors use sphingosine-phosphate receptor S1PR1 to probe the effect of ligand and receptor dimerization on its interactions with G proteins and b-arrestins. There are numerous issues with data presentation. In addition, the data are grossly over-interpreted. However, the topic is very interesting and potentially functionally important.

1. Lines 255, 523. Visual arrestin was called S-antigen at the time of discovery, and later called 48 kDa protein, arrestin, visual arrestin, rod arrestin, and arrestin-1, but it was never called S-arrestin. This needs to be corrected. Also, two systems of arrestin names are used, b-arrestins 1 and 2 are also called arrestin-2 and arrestin-3, respectively. The authors should provide the translation.
2. Lines 256-7. ERK binding site on b-arrestins was never elucidated. While some residues important for b-arrestin binding to upstream kinases Raf1 and MEK1 were identified, the interaction sites of these kinases on b-arrestins also remain unknown. Thus, there is no basis for the speculation in this area.
3. Figs. 1, 3, 5. The blots are overexposed. The data from at least three experiments should be statistically analyzed, and the results presented (as a bar graph or scatter plot).
4. Fig. 1 shows that PTX treatment eliminates both rapid and slow wave of ERK1/2 activation. This agrees with recent findings that G proteins, not arrestins, are required for GPCR signaling via ERK (J Biol Chem. 2016 Dec 30;291(53):27147-27159; Nat Commun. 2018 Jan 23;9(1):341; Sci Signal. 2017 Jun 20;10(484)) and does not agree with authors' interpretation. To show that b-arrestins are involved, the authors should have compared WT cells with b-arrestin1/2 knockout cells (described and used in all three references above).
5. Fig. 2 D, E and Suppl Fig. 5. The data suggest that each kind of S1PR1 receptor behaves in its own way and does not suggest S1PR1 interactions with mS1PR1 that the authors claim. Also, the authors do not present any evidence that the dimers they observed are parallel, rather than anti-parallel, while they interpret the results as if they know the orientation of the two receptors in these dimers.
6. Fig. 4B. "Two-tailed unpaired tests" mentioned on lines 519-520 imply Student's t-test. It is invalid when more than two groups are compared. ANOVA or equivalent with correction for multiple comparisons should be used.
7. Fig. 5B. Why only b-arr1 is shown? Quite a few papers claim that b-arr2 is the main player in ERK signaling.
8. Fig. 6. There is no structural evidence that arrestins or G proteins can bind GPCR dimers. In all crystal structures of the complexes (too numerous to reference here) a single molecule of arrestin or G protein binds a monomeric GPCRs. So, the figure is pure speculation. Moreover, in virtually all GPCR structures (with the sole exception of rhodopsin, including the second rendition of the arrestin-rhodopsin complex) the C-terminus was deleted or not visible. Besides, in many GPCRs the phosphorylation sites necessary for arrestin recruitment are not localized on the C-terminus, but on the i3 or other intracellular loops. The authors should correct the discussion and Fig. 6 accordingly.
9. Suppl Fig. 2. Two out of three blots overexposed to the point that they are unsuitable for quantification. ppERK should be quantified, the data from at least three experiments should be statistically analyzed, and the results presented (as a bar graph or scatter plot).

Reviewer #2 (Remarks to the Author):

Patrone et al. 2019, report an allosteric link between ligand binding, receptor oligomerization, and B-arrestin-binding at the sphingosine 1-phosphate receptor (S1PR1). Experimentally, this is done by comparison of the endogenous ligand, S1P, and the therapeutic ligand, FTY720-P, in pERK1/2 assays,

DLS experiments, ligand binding, IP pulldowns, and an internalization assay. From this, the authors conclude that the observed biased signaling is due to the oligomeric state of the receptors.

Overall, I think the authors have a very interesting set of data. It is seemingly well known that FTY720-P causes receptor internalization resulting in either functional antagonism or persistent signaling. Understanding this mechanism, and in part, if it is mediated via the oligomeric state of the receptor would be a major finding and of great interest. However, I think there are a few major issues with the reported work that prevent my recommendation for publication at this time.

Major concerns:

I am not an expert in the S1PR1 field, but there are quite a few publications regarding the pharmacology of FTY720-P and its effect on differentially modulating signaling pathways and S1PR1 internalization (a few examples: Healy et al BJP 2013 or Mullerhausen et al. Nat Chem Bio 2009). This work should be appropriately considered in the context of this manuscript and cited. These studies would also suggest that other signaling pathways beyond pERK are important. With that in mind, why was this study limited to mostly pERK?

A range of constructs are used in this study ranging from S1PR with a C-terminal eGFP fusion to a crystallization construct with an ICL3 T4L fusion. Figure 2A shows a clear effect on ligand binding with a C-tail truncation. What effect does an eGFP fusion have to the C-terminus? Likewise, what effect does T4L have on pERK1/2 signaling, presumably the receptor cannot signal. At no point is a truly WT construct used. I understand the technical reasons why the authors have chosen these constructs, however, is difficult to have a valid comparison across the different techniques with such differences in constructs. Radioligand binding studies and/or determination of S1P/FTY720-P potencies / efficacy would prove useful here.

For the DLS experiments, S1PR1 has a particle size or hydrodynamic radius of ~55nm. This seems incredibly large. That would suggest a particle diameter of >100 nm in size or 1,000 angstroms. Understandably, LMNG has a large and variable micelle size (Chaptal et. al 2017 Sci Rep), however even then when bound to a protein the expected diameter would be closer 5 nm. These values appear to be 10-fold off and cause great concern that the measured results are an artifact. How do these values change with DDM or DM? How do the samples look by size exclusion chromatography (SEC) preferably equipped with multi-angle light scattering (MALS)?

Ignoring the above issues with the DLS experiments, it is perhaps not to surprising that mutation of 3 residues could alter the oligomerization state of a receptor. This is after all a crystallization construct that was detergent purified and crystallized in a manner that suggests possible oligomerization. I am not sure that it would be appropriate to draw any other conclusions from these experiments than mutation of three residues (Y19A, F161A, and M180A) alter the oligomeric state of a detergent purified crystallization construct. And the addition of FTY720-P appears to make the non-mutated S1PR1 more polydisperse.

The authors next compare S1PR1 with mS1PR1 in a kinetic pERK1/2 experiment. Indeed, FTY-P appears to signal less, and importantly is not completely desensitized in the way that WT is. How do the authors know that these differences are not due to differences in the kinetics of S1P and FTY-P binding? From the S1P crystal structure, it is quite clear that the kinetics of binding (K_{on} and K_{off}) are going to be quite important for this receptor. Do any of the mutations effect the kinetics? They are certainly in a position of the receptor that could suggest they might. This needs to be tested as both single mutations and as the triple mutation. Additionally, what happens to S1P and FTY-P potency and efficacy in pERK? An appropriate review to consider for this manuscript: Lane et al 2017 Nat Chem

Biol, A kinetic view of GPCR allostery and biased agonism. Finally, and perhaps most importantly, what evidence is there that S1PR1 oligomerizes in cells? No proof or evidence is shown in this study, merely, just an extrapolation of results from the detergent purified system, which could be an artifact.

It is difficult to draw any convincing conclusions from the data in figure 4, and it likely has the same flaws as described above for figure 3. The results from figure 5 are interesting, particularly with regards to the loss in pre-coupling with arrestin, but again perhaps the basal state of the receptor is modified by the mutations (or even a mutation)? Overall, for the model reported in figure 6B there is simply not enough rigorous data to support any of the claims in this paper.

Minor concerns:

Overall, this manuscript is reasonably well written. However, there are a few times where word choice is confusing. For example, line 90, orthosteric site ligation? Do the authors mean binding? This is unclear. Other examples can be found. Likewise, the authors tend to overstate their claims, there is quite a bit of language that would need to be dialed down.

Do the authors really need to deplete their media of S1P via reverse dialysis? Has this been reported as an issue before?

Reviewer #3 (Remarks to the Author):

Patrone et al shows in an interesting manuscript that S1PR1 dimerization is differentially regulated by S1P and FTY720P and that this mechanism regulates β -arrestin complex formation and signaling of the receptor to the ERK pathway. Using modeling, mutagenesis, co-IP, endocytosis and signaling assays, the authors provide support for their novel interpretation of differences between physiological ligand S1P and the FDA-approved drug FTY720P. The concepts proposed are novel and potentially clinically relevant. Some aspects of this work are not as rigorous and needs additional evidence to fully support the conclusions.

Major issues:

1. When S1PR1 is endocytosed, particularly after FTY720P treatment, the receptor undergoes persistent phosphorylation, ubiquitinylation and degradation. The authors data do not show ANY receptor degradation, which is surprising. Since endocytosis and receptor degradation goes hand in hand, this needs to be resolved.
2. The key reagent developed by the authors, mutant S1PR1 in which three point mutations were introduced in the receptor dimer interphase is not well characterized. Does this construct induce Gi activation. ERK assay is not sufficient. The authors should examine direct G protein activation in their purified receptor preparations and in transfected cells.
3. CCG215022 is used at 50 μ M without any controls on target engagement, specificity and off target effects.
4. The internalization data (Figure 4) is not convincing. Co-localization experiments may be useful. However, receptor degradation experiments should be done to confirm this (as above).
5. Does the mutant receptor recycle after S1P was off?

6. Proper citation of receptor phosphorylation/ ubiquitylation/ degradation papers is needed.

Reviewer #1 (Remarks to the Author):

The authors use sphingosine-phosphate receptor S1PR1 to probe the effect of ligand and receptor dimerization on its interactions with G proteins and b-arrestins. There are numerous issues with data presentation. In addition, the data are grossly over-interpreted. However, the topic is very interesting and potentially functionally important.

We thank the reviewer for appreciating the interest of the results within the framework of the topic of GPCR signalling. We are confident that our response, stemming from new experiments and clarifications to some improperly presented issues, addresses the points raised by the reviewer.

1. Lines 255, 523. Visual arrestin was called S-antigen at the time of discovery, and later called 48 kDa protein, arrestin, visual arrestin, rod arrestin, and arrestin-1, but it was never called S-arrestin. This needs to be corrected. Also, two systems of arrestin names are used, b-arrestins 1 and 2 are also called arrestin-2 and arrestin-3, respectively. The authors should provide the translation.

We apologize for the confusion caused with our mixed usage of the nomenclature. The nomenclature dichotomy is now explained in the first instance, and now consistent throughout the text. The “S-arrestin” in complex with rhodopsin was corrected to “visual arrestin”.

2. Lines 256-7. ERK binding site on b-arrestins was never elucidated. While some residues important for b-arrestin binding to upstream kinases Raf1 and MEK1 were identified, the interaction sites of these kinases on b-arrestins also remain unknown. Thus, there is no basis for the speculation in this area.

We inferred the β -arrestin interaction site on ERK based on the results presented in the paper “Mutations of beta-arrestin2 that limit self-association also interfere with interactions with the beta2-adrenoceptor and the ERK1/2 MAPKs: implications for beta2-adrenoceptor signalling via the ERK1/2 MAPKs” by Xu et al. *Biochem J* 413, 51-60 (2008). The authors of the manuscript demonstrate via spot peptide arrays and FRET that residues at positions 285, 286 and 295 of β -arrestin1 are crucial for interactions with ERK1/2. This evidence was also used for the modelling of the structure of a possible β -arrestin signalosome complex in the paper “Unraveling the molecular architecture of a G protein-coupled receptor/ β -arrestin/Erk module complex” by Borquard et al. *Sci Reports* 5:10760 (2015). Thus, we believe that existing data support a possible interpretation of our results with a steric exclusion mechanism. We agree with the reviewer that the model we present is speculative, in the sense that we cannot (and do not) claim a high accuracy, and we added a statement (page 12) to avoid confusion to the reader. In the light of the reviewer’s concern, we emphasized the existing data on ERK/ β -arrestin interactions (refs. 37 and 38) and rephrased the discussion to further underscore that further structural analysis is required to obtain more detailed information (page 15).

3. Figs. 1, 3, 5. The blots are overexposed. The data from at least three experiments should be statistically analyzed, and the results presented (as a bar graph or scatter plot).

We agree with the reviewer on the importance of providing reproducible and statistically significant data. For this reason, we chose to take advantage of an ELISA assay for the quantification of ERK phosphorylation in all the kinetic experiments presented. The Western blots are provided as additional experimental evidence to show the variation (or lack thereof)

of cellular pERK, but were not used for quantitative purposes. Regarding the saturation of the images shown, all Western blot images were acquired using a ChemiDoc instrument (BioRad), coupled to a 6 megapixels CCD detector with a saturation of 65,536 counts and a nominal dynamic range > 4 orders of magnitude, as per manufacturer's specifications. In none of the images acquired detector saturation was reached. Together with the revised manuscript, we provide sample raw images from the Western blots presented in the Figures formerly numbered 1, 3 and 5 (now Supplementary Figures 2 and 4, and Figure 4) which can be viewed using the proprietary software ImageLab by BioRad, but also with other programs such as ImageJ, to ascertain that the maximum value recorded in the image is indeed below the detector saturation. For convenience, we here provide the histogram distribution of intensities in two raw images used to assemble Supplementary Figure 2 panel C (detection of total ERK and S1PR1) that shows no saturated pixels. We are willing to upload all the raw images presented in the manuscript to a server for accession, if this is deemed necessary. However, we would like to further reiterate that all the kinetics of pERK variation presented are derived from the ELISA experiments, and the Western blots are presented with the sole purpose of showing that indeed the absorbance values measured in the assay correlate with a variation in pERK.

Left: total ERK. Right: S1PR1-eGFP

4. Fig. 1 shows that PTX treatment eliminates both rapid and slow wave of ERK1/2 activation. This agrees with recent findings that G proteins, not arrestins, are required for GPCR signaling via ERK (*J Biol Chem.* 2016 Dec 30;291(53):27147-27159; *Nat Commun.* 2018 Jan 23;9(1):341; *Sci Signal.* 2017 Jun 20;10(484)) and does not agree with authors' interpretation. To show that *b*-arrestins are involved, the authors should have compared WT cells with *b*-arrestin1/2 knockout cells (described and used in all three references above).

We apologize with the reviewer for not being clear on these points, and for not adequately representing the complexity of GPCR-mediated intracellular signalling and the ongoing debate on the relative contributions by G-proteins and β -arrestins. Indeed, the papers pointed out by the reviewer sustain a primary G-protein involvement in ERK signalling. Other groups, however, showed that β -arrestin knockout using genetic manipulation with CRISPR/Cas9 results in a significant variability in the signalling outcome (*Sci Signal.* 2018 11(549) 7650). Thus, we believe this is a highly-debated topic that undoubtedly requires further studies. We rephrased the conclusions from the data shown in Figure 1. As the reviewer pointed out, the PTX treatment indeed shows a pivotal role of G-proteins in ERK activation (which we originally stated on page 6). The inhibition of GRKs, and thus the reduction of β -arrestin association to S1PR1, affects both the agonist efficacy and, at least for FTY720-P, the late phase response. At the same time, the co-immunoprecipitations we

performed on S1PR1 clearly show a ligand-dependent recruitment of β -arrestins and ERK at the receptor, that is one of our main conclusions. We clarified this in the text (pages 12 and 16) and added appropriate references (ref. 14) to the previous works analysing β -arrestin signalling in GPCRs, while avoiding any confusing statements on G α / β -arrestin involvement.

5. Fig. 2 D, E and Suppl Fig. 5. The data suggest that each kind of S1PR1 receptor behaves in its own way and does not suggest S1PR1 interactions with mS1PR1 that the authors claim. Also, the authors do not present any evidence that the dimers they observed are parallel, rather than anti-parallel, while they interpret the results as if they know the orientation of the two receptors in these dimers.

We added a more extensive description of the DLS experiments in the legend of Supplementary Figure 4. In Panel D, the total protein concentration is constant in each measurement (12.5 μ M), while the ratio between S1PR1 and mS1PR1 varies. Thus, the decrease of the population associated with the smaller radii (mS1PR1) is consistent with an interaction between mS1PR1 and S1PR1, resulting in mixed dimers of larger radii. We realize that these isolated experiments were a possible source of confusion, since they were carried out on recombinant, chimeric constructs. Thus, we performed co-immunoprecipitation experiments after co-transfecting the S1PR1-eGFP or mS1PR1-eGFP-expressing clones with either S1PR1 or mS1PR1. The results show that

- a) wild type S1PR1 interacts with itself;
- b) S1PR1 interacts with mS1PR1, as the DLS experiments suggested;
- c) mS1PR1 does not self-interact
- d) Neither S1PR1 nor mS1PR1 co-immunoprecipitate with the lysophosphatidic acid receptor, a lipid-binding GPCR that binds to structurally similar ligands.

The blots have been included in Figure 2, and provide conclusive evidence that the S1PR1 expressed in a biologically relevant system forms oligomers at the cell surface. We moved the DLS experiment as a Supplementary Figure, as it now constitutes a relevant proof that the recombinant constructs interact without requiring additional partners.

Regarding the orientation, the dimers observed in the truncated S1PR1 crystal structure that guided the mutational analysis performed, are parallel, and consistent with a possibly functional dimeric structure at the plasma membrane. The newly-added co-immunoprecipitation and mutagenesis studies are consistent with this arrangement. This is now stated on page 7.

6. Fig. 4B. “Two-tailed unpaired tests” mentioned on lines 519-520 imply Student’s t-test. It is invalid when more than two groups are compared. ANOVA or equivalent with correction for multiple comparisons should be used.

We apologize again for a poor wording of the statistical analysis performed. The test carried out is a Kruskal-Wallis, performed on the three conditions compared. This test is indeed an ANOVA test, that is non-parametric and allows comparison of multiple categories. The “two-tailed” definition refers to the assessment whether the distributions are shifted in one direction or the other, and “unpaired” implies that no reference distribution is used. Thus, we now state that the test carried out is an ANOVA test (page 29).

7. Fig. 5B. Why only b-arr1 is shown? Quite a few papers claim that b-arr2 is the main player in ERK signaling.

The reviewer correctly pointed out this issue that we initially excluded because of the striking result obtained with β -arrestin1. We now added a Western blot showing also the association

of β -arrestin2 to S1PR1 in Figure 4C. The results show that little β -arrestin2 is associated with the receptor at the basal level. Upon FTY720-P stimulation we detected a similarly large increase of receptor-associated β -arrestin2 after 30 minutes. As for β -arrestin1, S1P did not elicit tight association with β -arrestin2.

8. Fig. 6. There is no structural evidence that arrestins or G proteins can bind GPCR dimers. In all crystal structures of the complexes (too numerous to reference here) a single molecule of arrestin or G protein binds a monomeric GPCRs. So, the figure is pure speculation. Moreover, in virtually all GPCR structures (with the sole exception of rhodopsin, including the second rendition of the arrestin-rhodopsin complex) the C-terminus was deleted or not visible. Besides, in many GPCRs the phosphorylation sites necessary for arrestin recruitment are not localized on the C-terminus, but on the i3 or other intracellular loops. The authors should correct the discussion and Fig. 6 accordingly.

We are sorry for the confusing choice of words used in this section, as it was not our intention to state that all GPCRs may behave as apparently S1PR1 does. We agree with the reviewer on the state of the art of the structural analysis of GPCRs, bearing however in mind that GPCRs with proven dimeric structures (e.g. CXCR4) have not been crystallized in complex with either G-protein(s) or β -arrestin. Since S1PR1 was observed as a dimer in the crystals used in the x-ray analysis and our data also show the formation of S1PR1 oligomers via the same protein surface (α 4 helix), we attempted to model a higher order complex based on the existing structure of the S1PR1 dimer and the rhodopsin:arrestin complex. This model by no means represents an experimentally determined structure, but simply an aid to the visualization of the information deriving from the experiments here presented on S1PR1, with an eye to the effect of FTY720-P stimulation on ERK association to β -arrestin. We modified the figure of the model to include the phosphorylation of both the tail and the ICL3 upon agonist binding to S1PR1.

9. Suppl Fig. 2. Two out of three blots overexposed to the point that they are unsuitable for quantification. ppERK should be quantified, the data from at least three experiments should be statistically analyzed, and the results presented (as a bar graph or scatter plot).

Also in this case, the blots were provided to integrate the information from the pERK quantification measured with the ELISA assay. As previously stated, the digital images acquired do not show any saturated pixels. We include the original raw images for inspection, and a histogram analysis for the reviewer's convenience is here presented.

Reviewer #2 (Remarks to the Author):

Patrone et al. 2019, report an allosteric link between ligand binding, receptor oligomerization, and B-arrestin-binding at the sphingosine 1-phosphate receptor (S1PR1). Experimentally, this is done by comparison of the endogenous ligand, S1P, and the therapeutic ligand, FTY720-P, in pERK1/2 assays, DLS experiments, ligand binding, IP pulldowns, and an internalization assay. From this, the authors conclude that the observed biased signaling is due to the oligomeric state of the receptors.

Overall, I think the authors have a very interesting set of data. It is seemingly well known that FTY720-P causes receptor internalization resulting in either functional antagonism or persistent signaling. Understanding this mechanism, and in part, if it is mediated via the oligomeric state of the receptor would be a major finding and of great interest. However, I

think there are a few major issues with the reported work that prevent my recommendation for publication at this time.

Major concerns:

I am not an expert in the S1PR1 field, but there are quite a few publications regarding the pharmacology of FTY720-P and its effect on differentially modulating signaling pathways and S1PR1 internalization (a few examples: Healy et al BJP 2013 or Mullerhausen et al. Nat Chem Bio 2009). This work should be appropriately considered in the context of this manuscript and cited. These studies would also suggest that other signaling pathways beyond pERK are important. With that in mind, why was this study limited to mostly pERK? We acknowledge the importance of the papers pointed out by the reviewer, and appropriate references were added to the text (page 5, refs. 29 and 30). Indeed, several intracellular pathways are activated by S1PR1 triggering as we mentioned in the introduction. We focused on ERK activation because of the double valence of this mediator as a readout of both G-protein and β -arrestin-mediated GPCR signalling. We were interested in following a possible bias of the natural ligand and therapeutic agonist, and ERK phosphorylation is the main readout in the vast literature regarding GPCR signalling, allowing a straightforward comparison with all the other well-characterized receptors. Following the reviewer's suggestion, we added a reference to the papers the reviewer pointed out, to underscore that other signalling pathways may be affected by receptor oligomerisation (ref. 29 and 30), and that is an issue worth investigating further.

A range of constructs are used in this study ranging from S1PR with a C-terminal eGFP fusion to a crystallization construct with an ICL3 T4L fusion. Figure 2A shows a clear effect on ligand binding with a C-tail truncation. What effect does an eGFP fusion have to the C-terminus? Likewise, what effect does T4L have on pERK1/2 signaling, presumably the receptor cannot signal. At no point is a truly WT construct used. I understand the technical reasons why the authors have chosen these constructs, however, is difficult to have a valid comparison across the different techniques with such differences in constructs. Radioligand binding studies and/or determination of S1P/FTY720-P potencies / efficacy would prove useful here.

Indeed, we were forced to use a variety of S1PR1 constructs to investigate all the findings that arose from initial experiments, and we thank the reviewer for her/his understanding of the technical difficulties associated with the molecular and cellular characterization of GPCRs. That said:

1. We should clarify that the constructs used for obtaining soluble, recombinant protein for structural studies included T4L in place of ICL3 (Supplementary Figure 1). However, the constructs used in the cellular assays were "native" in their ICL3 and thus signalling capabilities.

2. Regarding the C-terminal eGFP fusion, we performed a new full set of signalling experiments using a truly unmodified S1PR1 and its triple mutant. We observed no variation in ERK1/2 phosphorylation induced by the C-terminal fusion of eGFP, confirming the suitability of using the fusion construct for the other functional experiments. The data are presented in Supplementary Figure 7.

3. We performed new experiments to determine the potency and the efficacy of the S1P and FTY720-P ligands on the eGFP-fusion receptors by following the increase in ERK phosphorylation 5 minutes after stimulus with different ligand concentrations. The titrations show equal response to S1P and FTY720-P in both the S1PR1 and mS1PR1-expressing

cells. We see a decrease in ligand potency with mS1PR1-expressing clones, but not in the efficacy (please note that all data were normalized to the highest response achieved using FTY720-P-stimulated S1PR1). These results are shown in Figure 2C.

For the DLS experiments, S1PR1 has a particle size or hydrodynamic radius of ~55nm. This seems incredibly large. That would suggest a particle diameter of >100 nm in size or 1,000 angstroms. Understandably, LMNG has a large and variable micelle size (Chaptal et. al 2017 Sci Rep), however even then when bound to a protein the expected diameter would be closer 5 nm. These values appear to be 10-fold off and cause great concern that the measured results are an artifact. How do these values change with DDM or DM? How do the samples look by size exclusion chromatography (SEC) preferably equipped with multi-angle light scattering (MALS)?

We agree with the reviewer that the radii derived from the DLS experiments were too large for the S1PR1 molecule, even in a dimeric arrangement, and we apologize for not adequately discussing the issue in the original manuscript. Following the reviewer's concerns, we invested a lot of effort to clarify the issue, and the results are here presented in detail, and summarized in the end:

1. The values of the hydrodynamic radii are obtained from the Stokes-Einstein equation $D = k_B T / (6 \pi \eta R_h)$, and we measured the viscosity of the detergent-containing buffer ($\eta_R = 1.5$). Thus, the values shown are now corrected for the difference in viscosity compared to PBS.

2. All recombinant proteins (S1PR1, its "monomeric" triple mutant, the mutant lacking the palmitoylation sites) elute as a retained, near-symmetrical single peak from SEC experiments using a Superdex200 column (separation range 10-600 kDa for globular proteins) at an elution volume of 10.32 mL, while mS1PR1 elutes at 10.61 mL. The void volume of the column where aggregated proteins elute corresponds to 8.2 mL, hence we are confident that the sample is not aggregated in a non-specific manner. Unfortunately, we could not gain access to a SEC-MALS instrument. Yet, the elution profile clearly shows that the samples are retained (presented in Supplementary Figure 4A).

3. As per reviewer's suggestion, we analysed the protein using a different set of detergents. First, we purified the protein using Cymal-5 (5-cyclohexyl-1-pentyl- β -D-maltoside) instead of LMNG (lauryl maltose neopentyl glycol). The protein behaved similarly in SEC experiments, and the DLS analysis revealed particles of similarly large size (Supplementary Figure 4D). To further investigate the issue, we exchanged the LMNG detergent with an amphipol. The amphipol-trapped purified receptors display a smaller particle size, both for the S1PR1-T4L (9.3 nm) and the mS1PR1-T4L (5.8 nm) constructs, values closer to what expected for a GPCR. For the sake of academic discussion, the large particles could be explained by large multimeric states of the recombinant constructs in solution when purified with LMNG or Cymal-5 that are prevented, at least in part, by the interactions of the receptor with the amphiphilic polymer used. Yet, we do not wish to speculate on this issue. Since other experiments (see point 5 below) corroborate the interaction of the receptors at the cell surface, we now present this extended DLS analysis as supplementary material.

4. The specific effect of FTY720-P binding to recombinant S1PR1 polydispersity were maintained regardless of the detergent or polymer used for the stabilisation of the receptor, and this effect was not seen on the mutated mS1PR1, or with other ligands (S1P) or analogues (non-phosphorylated FTY-720 or sphingosine). Thus, we feel that the variation in polydispersity is a ligand-specific effect that does not result from non-specific hydrophobic interactions, but rather orthosteric site occupancy. We agree with the reviewer that this finding was object of overinterpretation. We now modified the text to report the effect on the

particle size distribution, and do not link this to conformational variations or molecular flexibility (page 15).

5. To univocally demonstrate that in living cells S1PR1 forms oligomers, we resorted to co-immunoprecipitation experiments. When clones expressing S1PR1-eGFP were co-transfected with unmodified S1PR1, the two proteins were co-immunoprecipitated by an anti-eGFP nanobody. Instead, no co-immunoprecipitation was observed when mS1PR1-eGFP-expressing clones were transfected with mS1PR1. We also checked that the lysophosphatidic acid (LPA) receptor endogenously expressed in HEK293 cells did not co-immunoprecipitate with either S1PR1 or mS1PR1. The data from these new experiments provide strong evidence that S1PR1 oligomers are present at the plasma membrane, while the triple mutant affecting the surface of interaction observed in the truncated S1PR1 crystals does not.

6. As an added control, we also transfected S1PR1-eGFP-expressing cells with untagged mS1PR1, and showed that the two receptors co-immunoprecipitate. This result shows that the mutant protein not only retains the signalling capability (as shown in the kinetic experiments, Figure 3), but also a sufficient structural integrity to interact with the unmutated receptor.

In summary, following this extensive analysis, the following actions were taken:

- a. The co-immunoprecipitation results from the new experiments are now presented in Figure 2D to show receptor oligomerization in cells, and the lack of interaction between mS1PR1 molecules.
- b. The SEC analysis of the purified receptors is included as Supplementary Figure 4D.
- c. The DLS data, integrated with the new experiments performed on recombinant S1PR1s stabilized by different detergents, are now presented in Figure 2E and Supplementary Figure 4C-E. In the figure legend, we explicitly discuss the derived hydrodynamic radii.
- d. We rewrote the discussion of the DLS data to avoid over-interpretation regarding oligomeric state and conformational changes.

The co-IP data demonstrate the self-association of S1PR1 in cells, while the triple mutant loses this capability. These results parallel the observations with the purified, recombinant receptors, where mutations in the interaction surface result in decreased dimensions. We believe that these findings should address the concerns raised on the receptor oligomerization in a physiological setting.

Ignoring the above issues with the DLS experiments, it is perhaps not to surprising that mutation of 3 residues could alter the oligomerization state of a receptor. This is after all a crystallization construct that was detergent purified and crystallized in a manner that suggests possible oligomerization. I am not sure that it would be appropriate to draw any other conclusions from these experiments than mutation of three residues (Y19A, F161A, and M180A) alter the oligomeric state of a detergent purified crystallization construct. And the addition of FTY720-P appears to make the non-mutated S1PR1 more polydisperse.

We agree with the reviewer that we probably overstated the interpretation of the effect of FTY720-P on S1PR1 structure. Thus, we changed the text to reflect a more careful interpretation. Still, the effect of FTY720-P on detergent-purified S1PR1 is highly specific and, as shown above, it is present regardless of the detergent used in the purification of the recombinant receptor. We now point out the effect may reflect dynamics in quaternary structure, but speculate no further (page 15).

Regarding the oligomerization, we believe that the co-immunoprecipitation experiments now added to the manuscript strengthen the view that the mutation introduced in the mS1PR1 has not only an effect on the oligomerization state of the purified constructs, but also on the proteins when expressed in living cells.

The authors next compare S1PR1 with mS1PR1 in a kinetic pERK1/2 experiment. Indeed, FTY-P appears to signal less, and importantly is not completely desensitized in the way that WT is. How do the authors know that these differences are not due to differences in the kinetics of S1P and FTY-P binding? From the S1P crystal structure, it is quite clear that the kinetics of binding (K_{on} and K_{off}) are going to be quite important for this receptor. Do any of the mutations effect the kinetics? They are certainly in a position of the receptor that could suggest they might. This needs to be tested as both single mutations and as the triple mutation. Additionally, what happens to S1P and FTY-P potency and efficacy in pERK? An appropriate review to consider for this manuscript: Lane et al 2017 Nat Chem Biol, A kinetic view of GPCR allostery and biased agonism. Finally, and perhaps most importantly, what evidence is there that S1PR1 oligomerizes in cells? No proof or evidence is shown in this study, merely, just an extrapolation of results from the detergent purified system, which could be an artifact.

The central point raised by the reviewer is the lack of proof of S1PR1 oligomerisation in cells. We thus transfected S1PR1-eGFP- or mS1PR1-eGFP-expressing clones with unmodified S1PR1 or mS1PR1, and performed co-immunoprecipitation experiments using an anti-eGFP nanobody. The Western blots, shown in Figure 2, clearly show that S1PR1 self-associates. Instead, mS1PR1 does not form oligomers at the cell surface. Thus, these new experiments consolidate the pivotal point of our findings. We thank the reviewer for this suggestion, as it undoubtedly strengthened the manuscript.

Regarding the potency and efficacy variations upon introduction of the three amino acid substitutions, there is a decrease of both S1P and FTY720-P potency but not efficacy towards the mS1PR1. This is shown in Figure 2C.

We also agree with the reviewer that not only affinity but also binding kinetics could affect the signalling outcome. Unfortunately, the structure of S1PR1 bound to the antagonist ML056 is a static picture of a non-signalling receptor. Perhaps long molecular dynamics simulations may capture differences in kinetics of ligand binding, but at the present we do not know the structure of the activated receptor. Thus, from the isolated crystal structure we are unable to speculate on agonist binding kinetics. The mutations introduced are quite remote from the orthosteric site, but we cannot exclude that the allosteric capabilities of GPCRs may allow a long-range transmission from the mutation sites. Indeed, our results show a cross-talk between the tail, the monomer-monomer interface, and the orthosteric site. The potency and efficacy experiments now included, however, strongly advocate an affinity model rather than a kinetic one. In fact, the potency of the ligands is decreased for the mutated S1PR1, paralleling the lower affinities measured with the recombinant mS1PR1. However, the efficacy achieved at high ligand concentration is not affected by the mutations. Assuming a variation (increase) in k_{off} introduced by the mutations, one would expect a reduced efficacy.

Thus, the new immunoprecipitation and efficacy/potency experiments provide further support to our interpretation of the signalling data, and the effect of receptor oligomerisation.

It is difficult to draw any convincing conclusions from the data in figure 4, and it likely has the same flaws as described above for figure 3.

We believe that the further characterisation performed on the construct used, as shown in the previous point, now satisfy the reviewer's concerns.

The results from figure 5 are interesting, particularly with regards to the loss in pre-coupling with arrestin, but again perhaps the basal state of the receptor is modified by the mutations (or even a mutation)? Overall, for the model reported in figure 6B there is simply not enough rigorous data to support any of the claims in this paper.

The point raised by the reviewer is correct, and we now show the basal level of ERK phosphorylation is within experimental variation in the S1PR1 and mS1PR1-expressing clones (Figure 5B). Regarding the model presented in Figure 6B, we wished to provide a visual aid for the reader to wrap together the results. We do not claim that the figure represents the correct order of events leading to the FTY720-P effect, but is one possible way to visualize the steric exclusion model suggested by the data. We modified the figure to include more information on receptor phosphorylation events.

Minor concerns:

Overall, this manuscript is reasonably well written. However, there are a few times where word choice is confusing. For example, line 90, orthosteric site ligation? Do the authors mean binding? This is unclear. Other examples can be found. Likewise, the authors tend to overstate their claims, there is quite a bit of language that would need to be dialed down. We carefully revised the text to avoid any overinterpretation of the data, and added references to previous work to provide a more comprehensive depiction of the complexity of GPCR signalling.

Do the authors really need to deplete their media of S1P via reverse dialysis? Has this been reported as an issue before?

Yes, the process was important to avoid introducing biases when assaying the effect of S1P/FTY720-P on the S1PR1 receptor and its mutants. Both ligands require a carrier for the delivery to cells (albumin or HDL), and a S1P transporter-dependent bias could be observed when assaying S1PR1 signalling (J Biol Chem 287, 44645 (2012); Sci Signal 8, ra79 (2015); Nature 523, 342 (2015)). Serum deprivation is not an optimal option, as this would alter the albumin/HDL content. The removal of S1P from the media allowed the partition of the agonists in the two carriers without altering the carrier ratio (stated on page 5).

Reviewer #3 (Remarks to the Author):

Patrone et al shows in an interesting manuscript that S1PR1 dimerization is differentially regulated by S1P and FTY720P and that this mechanism regulates β -arrestin complex formation and signaling of the receptor to the ERK pathway. Using modeling, mutagenesis, co-IP, endocytosis and signaling assays, the authors provide support for their novel interpretation of differences between physiological ligand S1P and the FDA-approved drug FTY720P. The concepts proposed are novel and potentially clinically relevant.

Some aspects of this work are not as rigorous and needs additional evidence to fully support the conclusions.

We thank the reviewer for appreciating the novelty and implication of the results. We are confident that the new experiments performed and now included in the manuscript address the points raised, and further strengthen the rigor of the work performed.

Major issues:

1. *When S1PR1 is endocytosed, particularly after FTY720P treatment, the receptor undergoes persistent phosphorylation, ubiquitinylation and degradation. The authors data do not show ANY receptor degradation, which is surprising. Since endocytosis and receptor degradation goes hand in hand, this needs to be resolved.*

We performed a receptor degradation assay to provide further information on this issue. Specifically, we stimulated the S1PR1-eGFP and mS1PR1-eGFP-expressing clones with S1P and FTY720P for longer times and monitored residual eGFP fluorescence. Indeed, our initial analysis was limited to the time points required to follow ERK activation. As shown in Figure 4B, FTY720-P induced a markedly more pronounced receptor degradation, in line with what expected with the deep internalisation. S1P was not as effective, and neither ligand was as effective when the clones expressing the mutant receptor were analysed. Taken together, the receptor degradation data strengthen the view that FTY720-P has a specific effect on internalisation, and this effect is abrogated by mutations that prevent receptor association. We thank the reviewer for this suggestion!

2. *The key reagent developed by the authors, mutant S1PR1 in which three point mutations were introduced in the receptor dimer interphase is not well characterized. Does this construct induce Gi activation. ERK assay is not sufficient. The authors should examine direct G protein activation in their purified receptor preparations and in transfected cells.*

Unfortunately, the purified receptors carry a mutation (ICL3→ T4L) that prevents the binding of the G-proteins, thus cannot be assayed in vitro. The constructs used in the cellular assays are signalling competent, and we used the pertussis toxin (PTX) treatment on the mS1PR1-expressing clones to demonstrate the engagement of the G α_i protein for signalling. Addition of PTX, which specifically inactivates G $_i$ proteins, reduced the ERK phosphorylation to basal levels upon stimulation with either agonist. The results are summarized in Figure 3 and Supplementary Figure 7.

3. *CCG215022 is used at 50 μ M without any controls on target engagement, specificity and off target effects.*

We realize that the data originally presented were indeed heavily relying on previous literature, thus we performed and included the following analyses:

1. We demonstrated that the GRK inhibitor CCG215022 at 50 μ M does not inhibit EGF-mediated ERK1/2 phosphorylation (Supplementary Figure 2A, left panel);
2. We also verified that other S1PR1 downstream phosphorylation targets were not affected. We chose the AKT kinase as a reporter, since its phosphorylation is dependent on G α_i activity. Using an ELISA assay, we showed that AKT phosphorylation is G α_i dependent (since PTX treatment abolishes its modification), is induced by both S1P and FTY720-P, and was not modified by CCG215022 addition at 50 μ M concentration (Supplementary Figure 2A, right panel)

4. *The internalization data (Figure 4) is not convincing. Co-localization experiments may be useful. However, receptor degradation experiments should be done to confirm this (as above).*

We agree with the reviewer that many different strategies may be used to visualize and assess the degree of internalisation. Our approach of visualizing the dynamics of S1PR1-eGFP and its mutant in living cells using confocal fluorescence microscopy has already been

used, for instance in Oo ML et al, (2007) J Biol Chem 282:9082 (reference 35, but also see refs. 29 and 31). The observation of large dot-like structures trafficking from the plasma membrane to the intracellular compartment prompted us to develop an image analysis system that would allow an unsupervised identification of such structures, and their quantification. We feel that this approach could be used in analogous systems, and offers complementary tools to well-established immunofluorescence techniques. To further confirm our results, following the reviewer's suggestion, we performed receptor degradation experiments. We were pleased to see that the results (outlined in the previous point, presented in Figure 4B) parallel what obtained from the analysis of the confocal images, and the known S1PR1 degradation occurring after prolonged FTY720-P stimulation. Thus, the graph was included in Figure 4 to complement the analysis of the microscopy data.

5. Does the mutant receptor recycle after S1P was off?

The endogenous S1P lyase enzyme expressed in HEK293 cells is known to provide homeostatic intracellular S1P dephosphorylation (Johnson KR et al. (2003) J Biol Chem 278:34541), and its action allows the recycling of S1PR1 to the cell surface. Moreover, S1P stimulation leads largely to monoubiquitinylation of the receptor (Oo ML et al, (2007) J Biol Chem 282:9082). The internalization of mS1PR1 after S1P stimulation is not affected by the mutations, as judged from the visual inspection and software-assisted deconvolution of the confocal immunofluorescence images (Figure 4). The new experiments performed to assess receptor degradation after agonist stimulation demonstrate that the mutant receptor is only marginally degraded after binding to either S1P or FTY720-P. Thus, we can infer that upon S1P stimulation and the subsequent metabolic degradation of the ligand both S1PR1 and mS1PR1 recycle efficiently to the cell surface without reaching deep intracellular localisation. We added a comment in the discussion (page 9) and referenced the existing trafficking studies (refs. 29, 31 and 35).

6. Proper citation of receptor phosphorylation/ ubiquitinylation/ degradation papers is needed.

We added proper references to S1PR1 post-translational modifications, ubiquitinylation, degradation, and trafficking studies (ref. 35)

REVIEWERS' COMMENTS:

Reviewer #1 (Remarks to the Author):

The authors use sphingosine-phosphate receptor S1PR1 to probe the effect of ligand and receptor dimerization on its interactions with G proteins and b-arrestins. They designed oligomerization-deficient version of S1PR1 to address these issues.

The manuscript was greatly improved in revision. Added experimental data address most of the points raised by the reviewers. The manuscript is well written, although most parts are longer than they should be and can be shortened, further improving the manuscript.

Reviewer #2 (Remarks to the Author):

The authors (Patrone et al. 2019) have submitted a considerably revised manuscript that addresses my previous concerns of over-interpreted results by adding new data/experiments and dialing down some of the language used. I commend the authors for the work they did in addressing reviewer concerns. The results of this study are indeed interesting, and I look forward to seeing this work published in the future.

A few minor comments:

Line 234: "had a significant albeit limited effect", unless a statistical test was used it would be more appropriate to leave out "significant" as this implies a test was done.

SEC trace in figures S4A. Font sizes need to be bigger as it is difficult to read, same for the standard curve.

Reviewer #3 (Remarks to the Author):

The authors have addressed all of my comments fully. Both new experimental data and discussion edits have been provided. Overall, this work is of high quality and very novel. In addition, it provides an important conceptual advance in this medically relevant area.

We thank all reviewers for their critiques and suggestions that lead to an improved manuscript, and for finding the manuscript suitable for publication.

Reviewer #1 (Remarks to the Author):

The authors use sphingosine-phosphate receptor S1PR1 to probe the effect of ligand and receptor dimerization on its interactions with G proteins and b-arrestins. They designed oligomerization-deficient version of S1PR1 to address these issues.

The manuscript was greatly improved in revision. Added experimental data address most of the points raised by the reviewers. The manuscript is well written, although most parts are longer than they should be and can be shortened, further improving the manuscript.

We thank the reviewer for the input in the reviewing process, and the positive comments on the revised version.

Reviewer #2 (Remarks to the Author):

The authors (Patrone et al. 2019) have submitted a considerably revised manuscript that addresses my previous concerns of over-interpreted results by adding new data/experiments and dialing down some of the language used. I commend the authors for the work they did in addressing reviewer concerns. The results of this study are indeed interesting, and I look forward to seeing this work published in the future.

We thank the reviewer for acknowledging our effort, and for agreeing with our interpretation of the data.

A few minor comments:

1. Line 234: "had a significant albeit limited effect", unless a statistical test was used it would be more appropriate to leave out "significant" as this implies a test was done.

The sentence on line 234 was modified to "had a limited effect";

2. SEC trace in figures S4A. Font sizes need to be bigger as it is difficult to read, same for the standard curve.

We replaced the SEC traces and calibration curve with panels where thicker lines were used, and used a bigger font size.

Reviewer #3 (Remarks to the Author):

The authors have addressed all of my comments fully. Both new experimental data and discussion edits have been provided. Overall, this work is of high quality and very novel. In addition, it provides an important conceptual advance in this medically relevant area.

We thank the reviewer for appreciating our work and efforts to improve the manuscript.